# The Nexus Solutions Tool (NEST) v1.0: An open platform for optimizing multi-scale energy-water-land system transformations

Adriano Vinca[1,2], Simon Parkinson[1,2], Edward Byers[1], Peter Burek[1], Zarrar Khan[3], Volker Krey[1,4], Fabio A. Diuana[5,1], Yaoping Wang[1], Ansir Ilyas[6], Alexandre C. Köberle[7,5], Iain Staffell[8], Stefan Pfenninger[9], Abubakr Muhammad[6], Andrew Rowe[2], Roberto Schaeffer[5], Narasimha D. Rao[10,1], Yoshihide Wada[1,11], Ned Djilali[2], and Keywan Riahi[1,12]

[1]International Institute for Applied Systems Analysis, Austria
[2]Institute for Integrated Energy Systems, University of Victoria, Canada
[3]Joint Global Change Research Institute, Pacific Northwest National Laboratory, United States
[4]Dept. of Energy & Process Engineering, Norwegian University of Science and Technology, Norway
[5]Energy Planning Program, Federal University of Rio de Janeiro, Brazil
[6]Center for Water Informatics & Technology, Lahore University of Management Sciences, Pakistan
[7]Grantham Institute, Faculty of Natural Sciences, Imperial College London, United Kingdom
[8]Centre for Environmental Policy, Faculty of Natural Sciences, Imperial College London, United Kingdom
[9] Dept. of Environmental Systems Science, ETH Zurich, Switzerland
[10]School of Forestry and Environmental Studies, Yale University, United States
[11]Department of Physical Geography, Faculty of Geosciences, Utrecht University, The Netherlands
[12]Institute for Thermal Engineering, TU Graz, Austria

**Correspondence:** Adriano Vinca (vinca@iiasa.ac.at)

**Abstract.** The energy-water-land nexus represents a critical leverage future policies must draw upon to reduce trade-offs between sustainable development objectives. Yet, existing long-term planning tools do not provide the scope or level of integration across the nexus to unravel important development constraints. Moreover, existing tools and data are not always made openly available or are implemented across disparate modeling platforms that can be difficult to link directly with modern scientific computing tools and databases. In this paper, we present the Nexus Solutions Tool (NEST): a new open modeling platform that integrates multi-scale energy-water-land resource optimization with distributed hydrological modeling. The new approach provides insights into the vulnerability of water, energy and land resources to future socioeconomic and climatic change and how multi-sectoral policies, technological solutions and investments can improve the resilience and sustainability of transformation pathways while avoiding counterproductive interactions among sectors. NEST can be applied at different spatial and temporal resolutions, and is designed specifically to tap into the growing body of open access geospatial data available through national inventories and the earth system modeling community. A case study analysis of the Indus River Basin in South Asia demonstrates the capability of the model to capture important interlinkages across system transformation pathways towards the United Nations' Sustainable Development Goals, including the intersections between local and regional transboundary policies and incremental investment costs from rapidly increasing regional consumption projected over the coming decades.

## 1 Introduction

The United Nations' Sustainable Development Goals (SDGs) provide 17 broad targets and diverse indicators for guiding humanity and the environment towards prosperity. Many of the SDG indicators are interdependent, and thus implementation
strategies should be based on a broader systems perspective (Liu et al., 2015; Nilsson et al., 2016; McCollum et al., 2018). The concept of nexus thinking has gained traction, and is increasingly applied within the context of the linkages among energy, water and land (EWL) resources (Khan et al., 2017; Liu et al., 2018b; Albrecht et al., 2018). A nexus approach balances EWL interactions across multiple sectors and livelihoods to better understand the synergies and trade-offs associated with meeting future resource demands in a sustainable way (Bazilian et al., 2011; Biggs et al., 2015; Liu et al., 2018a).

A number of previous studies address nexus challenges with computational modeling. Generally, these studies demonstrate that co-optimization, in the sense that decisions for EWL sectors are made simultaneously and incorporate the interlinkages, can identify strategies that avoid trade-offs and achieve synergies (Buras, 1979; Lall and Mays, 1981; Matsumoto and Mays, 1983; Huang et al., 2017; Kernan et al., 2017; Santhosh et al., 2014; Pereira-Cardenal et al., 2016; Dodder et al., 2016; Oikonomou and Parvania, 2018). Similarly, previous analyses integrated water, energy and food systems across multiple temporal and
spatial scales, and quantified the economic benefits that joint water-energy planning can provide by reducing the investment and operational costs of future infrastructure systems (Howells et al., 2013; Dubreuil et al., 2013; Parkinson et al., 2016; Zhang and Vesselinov, 2017; Khan et al., 2018; Bieber et al., 2018; Wang et al., 2018; Li et al., 2019; Vakilifard et al., 2019). Land-use impacts of energy decisions, including bioenergy supply-chain interactions, are also increasingly integrated into long-term energy planning models to provide improved estimates of biomass availability and cost (Mesfun et al., 2018; Akhtari et al.,
2018; de Carvalho Köberle, 2018). Analysis of decarbonization pathways for the United States demonstrates that multi-scale modeling is crucial for assessing the EWL nexus because of the diverse constraints on EWL resources at high spatial resolution, and the interaction with policies impacting different sectors and administrative levels (Sattler et al., 2012; Hejazi et al., 2015). Similarly, other recent global analysis with an integrated assessment model highlights important differences between spatial scales relevant for energy, water and food supply (Bijl et al., 2018). In this context, some large-scale hydro-economic and
integrated assessment models increasingly take a multi-scale perspective and consider water infrastructure investments across multiple basins, sectors and end-uses (Kahil et al., 2018; Robinson et al., 2015; Kim et al., 2016; Parkinson et al., 2019).

    Energy, water and land systems optimization models are thus important sectoral tools that inform utility and national planning towards the sustainable long-term development of natural resources. Yet, models used to develop long-term pathways consistent with the fundamental transformational changes called for under the SDGs do not represent simultaneously re-allocation
of resources and capacity expansion decisions across tightly linked EWL sectors. Cross-sector interactions are crucial to consider when resource availability is limited and infrastructure expansion is expensive. Unforeseen constraints could lead to stranded assets and vulnerable water, food and energy supplies. More integration across EWL systems and resource planning

decisions is required to capture important interactions in an explicit way, so that least-cost nexus solutions can be identified using engineering-economic tools such as optimization.

Leveraging open source tools will promote end-user accessibility and should be prioritized for long-term system optimization models to enable validation and re-use in future research (Howells et al., 2011; DeCarolis et al., 2017). Previous analysis combines different energy, water and land sector planning tools to achieve open-access integration (Welsch et al., 2014). The results of each sectoral planning tool are passed between tools as boundary conditions until the models reach an acceptable level of convergence. This process can take time and the decision solution obtained is not necessarily optimal across sectors. Moreover, the individual resource planning models require specific expertise to develop and run, and it can be time-consuming to design and implement a robust database for the model inputs and results, as well as online systems for sharing and merging model changes across different users. Other recent model developments are focusing mainly on water infrastructure (Payet-Burin et al., 2019) or city-scale scenarios (Bieber et al., 2018; McManamay et al., 2019), leaving room for improvement in terms of the sectoral and geographic scope for solutions.

In this paper, we present the NExus Solutions Tool (NEST): a new open platform for integrated EWL systems analysis under global change. The framework links a high-resolution distributed hydrological model to an engineering-economic modeling scheme that integrates multi-scale decisions impacting long-term EWL transformations. We mapped the output variables from NEST to the SDG indicators enabling integrated modeling of coordinated implementation and quantification of the investment costs. The new decision-making and open modeling platform provides a flexible framework for identifying and assessing EWL nexus solutions that can be applied to different geographic regions and multiple spatial and temporal scales.

The following section describes the NEST implementation. Sections 3 and 4 demonstrate the enhanced approach using data collected and processed for integrated policy analysis and capacity building in the Indus River Basin. Section 5 presents the conclusions and opportunities for future research.

## 2  Modeling framework

NEST links databases, processing scripts and state-of-the-art models covering multiple disciplines (Figure 1). The core framework consists of a distributed hydrological model (CWatM) and a resource supply planning model (MESSAGEix), both capturing the historical period and a future time horizon. NEST is used to generate future scenarios, where a scenario represents the technological and earth system transformation pathway under a given set of input data assumptions. In this context, the Shared Socioeconomic Pathways (SSP) and Representative Concentration Pathways (RCP) act as coupled scenario narratives framing climate and human development trajectories and driving exogenous demand profiles for specific sectors (Van Vuuren et al., 2011; O'Neill et al., 2017). Sectoral coverage is harmonized between CWatM and MESSAGEix so that demand profiles can be translated between models. CWatM is initially run under baseline conditions to inform MESSAGEix of dynamic constraints on water availability, hydropower potential and irrigation water requirements. In future work elements of the resulting MESSAGEix pathway will be passed back to CWatM to simulate the expected human impacts under adaptive management at a high spatial resolution (Figure 1).

## 2.1 Community Water Model (CWatM)

The Community Water Model (CWatM) provides a grid-based representation of terrestrial hydrology, applied in this instance at a spatial resolution of 5 arc-minutes (grid-cells approximately 8 km wide near the equator) and daily temporal resolution (Burek et al., 2019). CWatM distinguishes between six land cover types, including forest, irrigated non-paddy cropland, irrigated rice paddy, impervious surface, water bodies, and other land cover in simulating the water balance of each grid cell. CWatM includes processes relevant for high altitude implementations, including snow, glacier, and permafrost. Potential evaporation is calculated using the Penman-Monteith equations. Processes within soil layers include frost, infiltration, preferential flow, capillary rise, surface runoff, interflow, and groundwater percolation.

The model simulates river streamflow using the kinematic wave routing approach, and can simulate either naturalized streamflow or streamflow impacted by human activities including reservoirs, irrigation demand, and water withdrawals and return flows by industrial and domestic sectors. Reservoir outflow in the model is a function of the relative filling of the reservoir, storage parameters, and outflow parameters (Burek et al., 2013). Irrigation demand is a function of crop water demand, water availability, and crop type (paddy or non-paddy) (Wada et al., 2014). Parameter calibration uses an evolutionary algorithm that optimizes a modified version of the Kling-Gupta Efficiency (KGE) between the simulated and observed discharge (Fortin et al., 2012; Beck et al., 2016).

## 2.2 MESSAGEix

MESSAGEix is an open-source, dynamic systems-optimization model developed for strategic energy planning (Huppmann et al., 2019). MESSAGEix is based on the original MESSAGE model that has been developed and applied widely over the past three decades to analyze scenarios of energy system transformation, both globally and in different geographic regions, under technical-engineering constraints and political-societal considerations, e.g., (Messner and Strubegger, 1995; Riahi et al., 2007; Van Vliet et al., 2012; Kiani et al., 2013). A defining feature of MESSAGEix that distinguishes it from other energy models in its class (e.g., OSeMOSYS (Howells et al., 2011) and MARKAL (Loulou et al., 2004)), and that leverages its widespread use as a nexus solutions tool, is that it incorporates the ix modeling platform (ixmp): a back-end database and version control system that enables users to collaboratively develop, solve and visualize models using the open-source R and Python programming environments (Huppmann et al., 2019). This feature complements with the philosophy and design of CWatM, which utilizes similar open access software (Python) as the main interface for collaborative model development and calibration. In this context, the NEST framework employs the *reticulate* package to integrate R and python work environments (Ushey et al., 2019).

As a bottom-up systems optimization model, MESSAGEix includes resource consumption and capacity limitations at the technology-level. Each technology modeled in MESSAGEix is defined and characterized by input/output efficiencies (the rate at which a particular commodity is consumed or produced during technology operation), economic costs (investment, fixed and variable components), and environmental impacts (e.g., greenhouse gas emissions, water consumption, etc.). A technology in this context can represent any process that transfers or transforms commodities, including natural systems such as rivers,

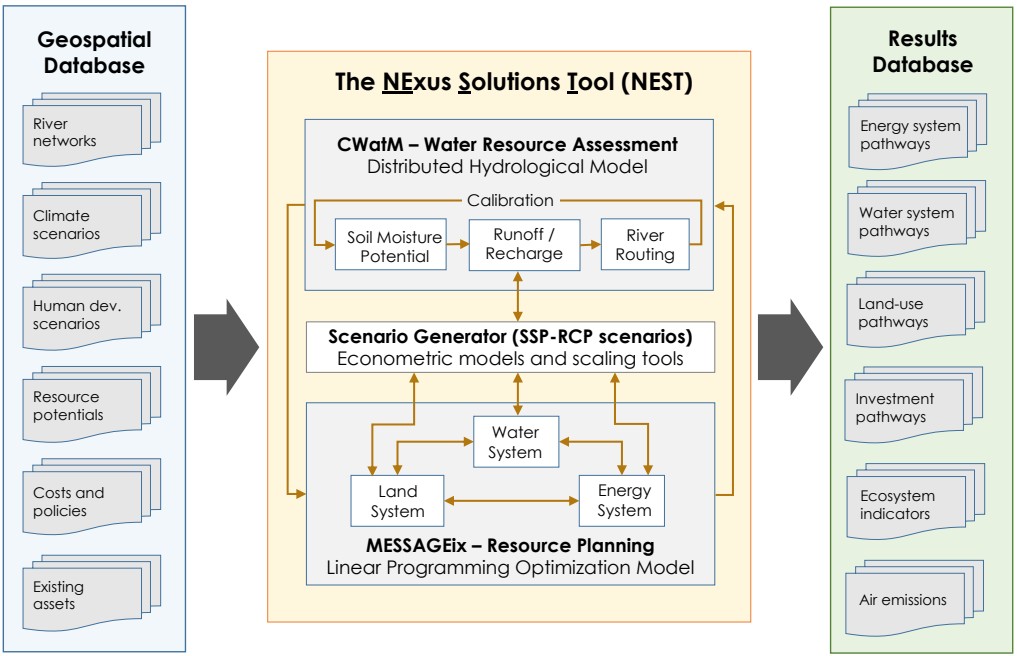

**Figure 1.** The NEST modeling scheme integrates the Community Water Model (CWatM) with a unified EWL technological system implemented in MESSAGEix. A scenario generator harmonizes data across the models and generates exogenous demand profiles aligned with coupled climate-human development narratives from the Shared Socioeconomic Pathways (SSP) and Representative Concentration Pathways (RCP).

aquifers and crops. By solving the following deterministic inter-regional and inter-temporal linear programming (LP) problem, MESSAGEix minimizes the total cost for system capacity and operation over a future time period while meeting user-specified levels of demand and technical/policy constraints:

$$\min f(\mathbf{x}) = \sum_{r,t} \mathbf{c}_{r,t}^{\mathrm{T}} \mathbf{x}_{r,t}\, \delta_{r,t}\; ; \; \mathbf{A}\,\mathbf{x} \geq \mathbf{b} \tag{1}$$

5   In the above system of equations, the time period index is given by $t$ and the region index is given by $r$. The solution vector containing the capacity and activity of the technologies is given by $\mathbf{x}$. Economic costs are described in the cost coefficient vector of the objective function $\mathbf{c}$. The discount rate associated with future cash flows is represented by $\delta$. The set of constraints including the supply-demand balances, capacity limits, technology retirements and capacity additions, activity bounds and additional policies addressing environmental impacts are contained in the technical coefficient matrix $\mathbf{A}$ and right-hand side

10   constraints vector $\mathbf{b}$. The full set of equations is summarized in the online model documentation (https://messageix.iiasa.ac.at). The single-objective LP formulation can also readily be transformed to handle multiple objectives, such as minimum total investments, emission level or other environmental indicators (Parkinson et al., 2018).

By linking the inputs and outputs of individual processes, energy, water and land decisions can be represented as a single system using the MESSAGEix modeling scheme. Thus, decisions impacting system design and operation over the planning horizon are made understanding the nexus interactions, and will adapt the transformation pathways for each sector to avoid constraints and reduce trade-offs from the perspective of the objective function. Moreover, MESSAGEix supports spatially-distributed systems modeling using a node-link representation, where commodities can be transferred between nodes based on the definition of dedicated technologies. It is therefore possible to explicitly represent the interplays between up- and downstream water users. Commodities are distinguished by the location (level) within the supply-chain enabling explicit accounting of associated efficiency losses and costs for grid and conveyance infrastructures. The temporal representation enables users to select the investment periods (e.g., annual) and sub-investment periods (e.g., sub-annual) over which supply, demand and system capacity must be balanced (Huppmann et al., 2019).

## 2.3 Reference system

The reference system is the user-defined bottom-up representation of the technological system and its spatio-temporal delineation in MESSAGEix that defines interactions between technologies and the balance of commodity flows across the system (Messner and Strubegger, 1995). The reference system contains the portfolio of possible technologies and interventions (existing and future) and does not typically change across scenarios; the parameterization of data, including the constraints, are varied to compare how the system reacts to certain inputs, policies and objectives. Two broad categories of data are used to characterize the NEST reference system: (1) historical data on resource use and availability, existing and planned technology capacities; and (2) parametric data for technologies used in the optimization model, expressed as costs or consumption of resources per unit of production. These data are based on assumptions and can vary spatially and temporally.

### 2.3.1 Spatial delineation

River basins are defined as fundamental spatial units in the reference system because they indicate how surface runoff (discharge) is directed across space and towards a single outlet downstream to the sea or an inland lake. River basins are disaggregated into sub-basins (tributaries) in NEST to improve accounting of within-basin surface water flows and impacts of upstream water use on downstream water availability. To enable a transboundary perspective, the approach further intersects the sub-basin boundaries with country administrative units; sub-national administrative units and regions covering multiple basins could be considered. The framework does not represent countries entirely, unless countries completely fit within the basin delineation or multiple basins covering a country's borders are included. Optionally, the units can be further intersected with agro-ecological zone boundaries to support diverse climatic characteristics within each sub-basin. The intersection of the administrative, agro-ecological and sub-basin units results in a new classification of management units defined as Basin Country Units (BCUs) (Figure 2) (Gaupp et al., 2015).

Each BCU is defined as a management unit (or node) in MESSAGEix. The nodes are an aggregated representation of the embedded resources and infrastructure assets that supply demands in the model, and are the fundamental spatial scale over which supply and demand are balanced. Infrastructure connections that move resources outside the region are included as

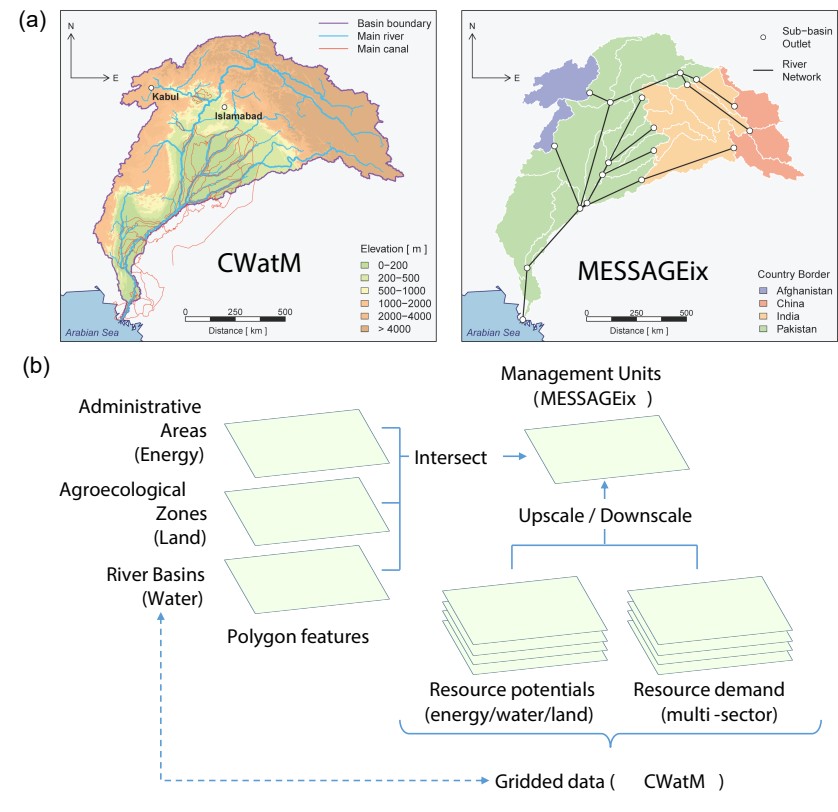

**Figure 2.** Delineation of spatial units in NEST. (a) The Indus River Basin elevation changes, river system in CWatM (left), and the basin delineated into Basin Country Units (BCUs) with a reduced form node-link river network for implementation in MESSAGEix (right); (b) the methodology intersecting basin boundaries, administrative regions and agro-ecological zones to converge on a common spatial scale, and linkage to gridded resource potentials and demands.

boundary conditions. A reduced-form network for guiding surface water flows between the BCUs is derived based on high-resolution flow-direction data consistent with CWatM (Kahil et al., 2018). An example for the Indus River Basin in South Asia is depicted in Figure (2). The approach is leveraging flow direction data at 15 arc-seconds from HydroSHEDS, which provides hydrographic data layers that allow for the derivation of watershed boundaries for any given location based on the
5    high-resolution Shuttle Radar Topography Mission (SRTM) digital elevation model (Lehner and Grill, 2013).

   Renewable surface water and groundwater inflows into each BCU are represented by aggregating (upscaling) the gridded run-off and recharge projections simulated with CWatM under current land-use patterns (Figure 2). This approach is likely to overestimate the available freshwater for human use within each BCU, because water users are distributed and do not have uniform access to the aggregate BCU-level water resources. Grid-cells in CWatM are mapped to specific management units
10   in MESSAGEix by overlaying the polygons and identifying the grid-cell centroids that fall within a given polygon boundary. Daily run-off sequences from CWatM are converted to decadal inflow scenarios by averaging monthly volumes over a 30-year

time period; inflow percentiles can alternatively be stipulated to consider extreme flow conditions. Similarly, other gridded resource potential and demand projections detailed in the following subsections are harmonized to the CWatM spatial grid to facilitate upscaling and downscaling between models.

### 2.3.2 Water sector

The scheme for water management within each BCU is depicted in Figure 3. Different water resources (surface, aquifer and saline) are accounted for and allocated across sectors (urban, rural, energy and agriculture). Internal runoff, regulation of reservoirs and water flowing from adjacent nodes through rivers or canals, all contribute to available surface water in each BCU. Renewable and non-renewable groundwater use is distinguished using groundwater recharge scenarios from CWatM and the efficiency losses from irrigation (Yang et al., 2016). Simultaneously, return-flow volumes are managed, including

opportunities to recycle wastewater streams within and between sectors. River flow and conveyance technologies move water between BCUs. Sectoral water withdrawals and return flows occurring outside the energy and land systems (i.e., municipal and manufacturing sectors) are exogenous and, together with endogenous water requirements for power plants and crops, drive the investments in water distribution and wastewater treatment infrastructure. Interactions across sectors are included in the model decision-making, including the energy required for pumping and treating water, and the water needed for crops and

electricity generation. Average elevation changes between major urban areas are used to estimate energy intensities for specific conveyance routes (Parkinson et al., 2016), whereas average water table depths are used to estimate energy intensities for lifting groundwater to the surface (Kahil et al., 2018).

Figure 3 depicts an explicit linkage enabled between nodal outflows and the production of hydropower potential in the model. The potential is passed to the energy system representation described in the following section, and limits the maximum monthly

hydropower generation in each BCU. An important challenge surrounds the aggregation of distributed hydropower potential that varies within each BCU both spatially and temporally. We estimate a linear transformation coefficient between modeled flows in the reduced-form basin network and the BCU-level hydropower potential calculated using the gridded data from the hydrological model. Hydropower projects off the main river tributary do not depend on upstream flows in the BCU river network and are identified based on the gridded flow direction data. Separate technologies and linear transformation coefficients

are defined for these projects, where the linear transformation coefficient is estimated using the internal BCU runoff. In NEST, we map the CWatM runoff data onto the 15 arc seconds flow accumulation grids from HydroSHEDS to estimate discharge at scales that preserve elevation differences governing hydropower potential (Gernaat et al., 2017; Korkovelos et al., 2018). Potential hydropower capacity $hp$ is calculated with the following equation:

$$hp = \eta \cdot \rho \cdot g \cdot q \cdot (h_i - h_o) \tag{2}$$

where $\eta$ is the turbine efficiency, $\rho$ is the density of water, $g$ is the gravitational acceleration, $q$ is the design discharge (taken to be the 70th percentile of the inflow sequence), $h_o$ is the outlet elevation and $h_i$ is the inlet elevation. Individual projects are identified along 5 km reaches of the 15-arc second river system based on their estimated annual production level and a set of exclusion zones including the distance to existing infrastructure, land-use and population density (Gernaat et al., 2017). We

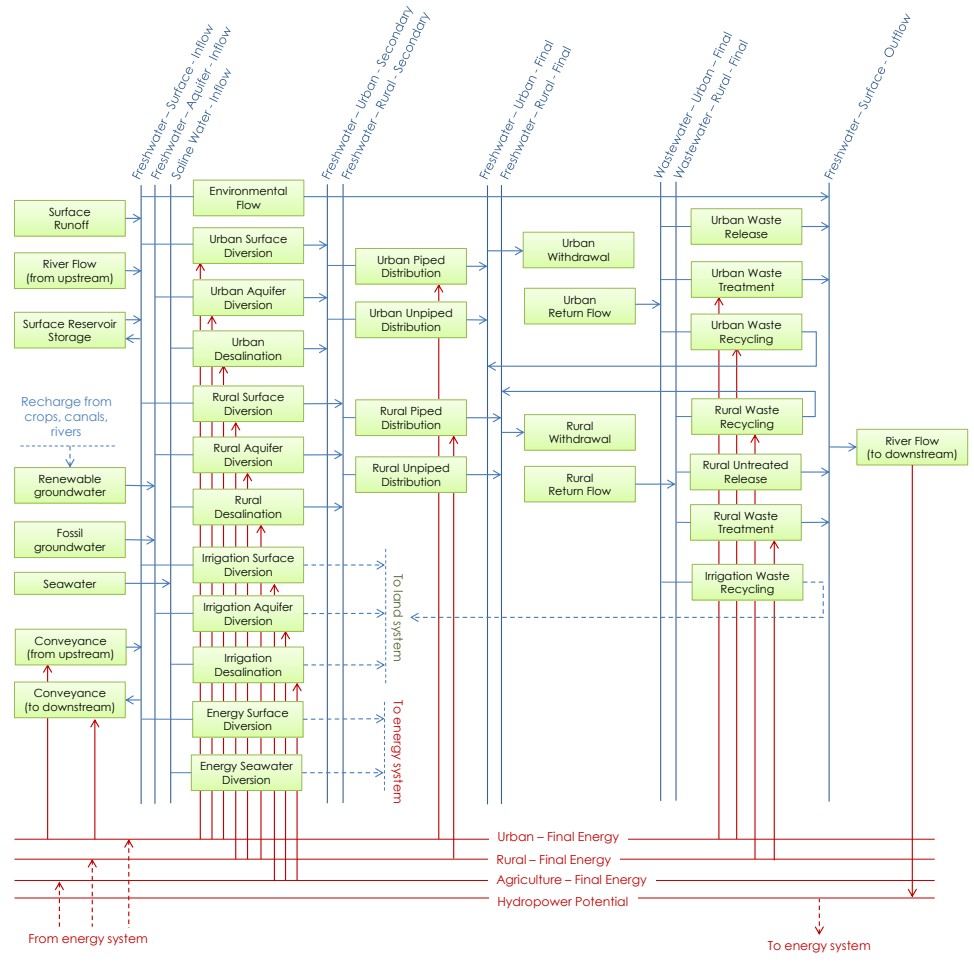

**Figure 3.** The water sector representation in each BCU using the MESSAGEix reference system scheme. The water sector is hard-linked to the energy and land sector representations using the indicated interactions.

assume that new projects can only utilize 10% of the total flow to ensure a high level of ecological security (Richter et al., 2012), and limit the canal lengths to a maximum of 3 km based on observed historical trends in installations. We do not consider dam storage or transfers of water between rivers in the assessment of hydropower potential due to additional planning challenges that are associated with these projects not readily monetized in the framework. Alternatively, new dam projects are considered

on a case-by-case basis based on published information on planned projects and stakeholder engagement.

### 2.3.3  Energy sector

The energy system representation for EWL nexus analysis using the MESSAGEix framework is depicted in Figure 4. The approach mimics closely conventional energy systems modeling with MESSAGEix, but integrates directly interactions with the novel implementation of the water and land systems. A diverse range of fossil and low-carbon energy resource extraction,

processing and power generation technologies can be included in the framework. Water system interactions are enabled through the definition of water withdrawal and consumption intensities for each energy technology and connection to water diversion technologies constrained by the availability of water resources. Thermal power plants are also distinguished by cooling technology, with the choice of cooling technology impacting the plant's economics and efficiencies. Alternative formulations may disaggregate the cooling technology choice from the prime mover technology in order to enable retro-fitting of cooling systems

directly (Parkinson et al., 2019).

Wind and solar potential is estimated by linking NEST to the Renewables.ninja application programming interface (https://www.renewables.ninja/). Renewables.ninja estimates hourly capacity factors for wind and solar technologies covering most terrestrial locations in the world, and generated based on calibrated resource data and technology representations (Pfenninger and Staffell, 2016; Staffell and Pfenninger, 2016). In NEST, the grid-cell centroids from CWatM are passed to Renewables.ninja

which then generates hourly production times series at each location. Exclusions zones are used to limit the areas where wind and solar can expand. The gridded potential in each management unit is categorized into capacity factors for representing diverse performance characteristics within each BCU.

A simple energy transfer scheme is considered for electricity transmission between adjacent BCUs, with distinct costs for each route estimated based on the average distances between the most populated urban area within each BCU (Parkinson

et al., 2016). Fuel trade with areas outside the delineated study region are defined using consistent fuel price projections from the MESSAGEix-GLOBIOM global integrated assessment model (Fricko et al., 2017). The energy system interacts directly with agriculture systems through the inclusion of bioenergy technologies that consume crop yields. Included are categories of dedicated bioenergy power plants providing electricity to grid-connected and distributed consumers, as well as categories for existing plant-types that can be co-fired using a limited fraction of bioenergy feedstock (e.g., crop residues). The current version

of the model does not account for the direct land footprint of energy system technologies. Energy demands from the agriculture and water sector activities are accounted for to ensure sufficient power generation capacity and to account for associated air pollution and greenhouse gas emissions.

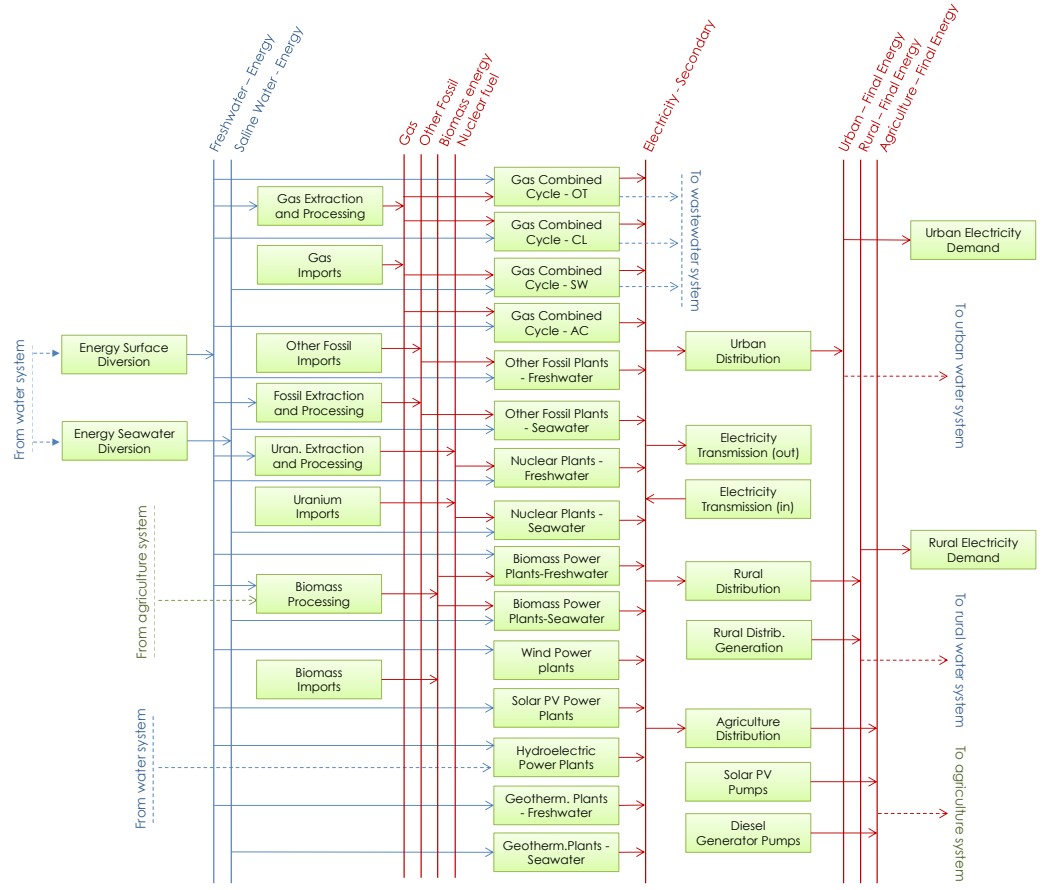

**Figure 4.** The energy sector representation in each BCU using the MESSAGEix reference system scheme. The energy sector is hard-linked to the water and agriculture sector representations using the indicated interactions.

### 2.3.4 Agriculture sector

An important feature of the reference system that bridges decision-making across the nexus is an agriculture sector representation integrated with the water and energy sectors presented previously. Diverse crop types and management strategies can be included in the approach, with the model selecting the cropping area and management method. The latter enables representation of alternative irrigation technologies, land preparation methods, and/or fertilizer application intensities, and importantly incorporates the spatial re-distribution of crops as a management strategy. We adopt a similar approach for integrating land-use into the reference system to that proposed in Köberle (2018) (de Carvalho Köberle, 2018), so that when the model selects a specific land-use it must balance the decision with the available land area within each BCU. Land-use is categorized into specific types (forest, pasture, crop, natural, etc.), with dedicated land-use change processes defined in the reference system

to convert land-use between types. The maximum cropping area is constrained based on the suitability of land within each BCU to support specific crop-types due to topographic and climatic conditions, as well as the total area available for cropping across all crop types. Non-CO$_2$ emissions as well as on-farm energy requirements besides that used for water pumping are tracked for different crops based on data from the literature (Rao et al., 2019). The model does not currently include dynamic growth and harvest of short-rotation forest crops, but this feature could be added in future work through appropriate definition in MESSAGEix using, e.g., the interannual stock and storage variables (Section 2.4). In Figure 5 we show an example for a system containing rice and wheat crop types with rain-fed, canal and drip irrigation options.

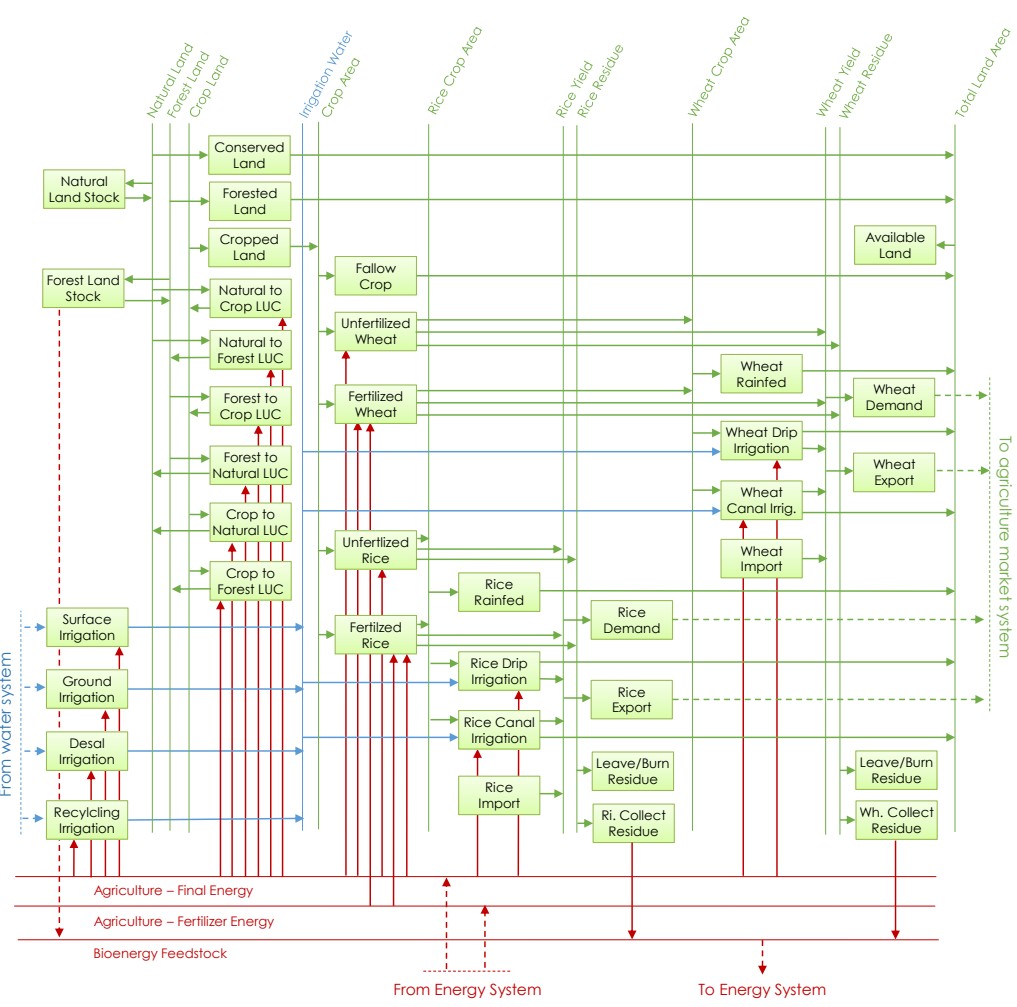

**Figure 5.** The agriculture sector representation in each BCU using the MESSAGEix reference system scheme. The agriculture sector is hard-linked to the water and energy sector representations using the indicated interactions.

For crop process modeling, each BCU aggregates crop parameters into coarser spatial units with average land-use parameters (Havlík et al., 2011). Crop yields are calculated aggregating spatial historical data at the BCU-level. This results in different yield coefficients for each crop, unit area and water supply (irrigation or rain). Similarly, crop water requirements vary across types and the intensity per unit area is estimated for each BCU using consistent water resource projections from the hydrological model. The irrigation per unit area for each crop $w$ is calculated using the CROPWAT approach (Smith, 1992):

$$w = \max \left\{ \left( k \cdot e - p^* \right), 0 \right\} \tag{3}$$

In the above equation, $k$ is the crop coefficient, $e$ is the reference evapotranspiration and $p^*$ is the effective precipitation. The reference evapotranspiration is calculated with CWatM using the Penman-Monteith method. The effective precipitation accounts for soil water storage and is estimated following the CROPWAT approach: (Smith, 1992):

$$p^* = \begin{cases} p \cdot (4.17 - 0.2 \cdot p) & p < 8.3 \text{mm/day} \\ 4.17 + 0.1 \cdot p & \text{otherwise} \end{cases} \tag{4}$$

For non-paddy crops, $p$ is the 10-day moving average daily precipitation (in mm/day), and for paddy crops it is the 3-day moving average to account for saturated soils (Döll, 2002). Irrigation intensities can optionally be calibrated such that, when aggregated across a given BCU, reproduce annual historical irrigation withdrawals when multiplied by the historical cropping area.

Similarly to the other sectors, the model defines the infrastructure portfolio to meet an exogenous demand for crop yields. Additionally to internal production, import and export of crop yields are allowed and demands can be defined and aggregated across multiple regions to simulate national accounts. Moreover, crop residues are tracked as by-products of agriculture activities. The residues can be burnt resulting in air emissions or transported and processed to have solid or liquid biofuel for electricity production.

### 2.3.5 Multi-sector demands and return-flows

Despite the endogenous representation of interactions between energy, water and land systems, there remains the need to exogenously define consumption profiles for the different sectors of the economy categorized in NEST but not specifically modeled at the technology-level. This currently includes the municipal and manufacturing sectors. Baseline demands for freshwater and cropping pattern are also required for the hydrological modeling. A demand scenario generator incorporated into NEST combines gridded climate and socioeconomic data from the coupled SSP-RCP scenario framework with econometric models fit to historical data. The SSP-RCP scenario data is harmonized at 7.5 arc-minutes and includes urban and rural populations, income-level and climatic indicators. Sector specific econometric models convert the gridded demand drivers into consumption profiles (water and electricity) and water infrastructure access rates for each sector (Parkinson et al., 2016, 2019). For regions lacking universal electricity access and transmission data, gridded electrification rates are estimated using satellite derived night-time light intensity combined with recent maps of population changes (Falchetta et al., 2019), and used to parameterize historical grid capacities and downscale national electricity projections from econometric models.

Food and fiber demands are represented as constraints on yields from specific crops aggregated to the national-scale. Import and export demands are included using variable prices, which might be calibrated in future work by optimizing parameter settings so that the model is able to reproduce prices observed historically (Howitt, 1995). Transport of agricultural products is not considered in the modeling, but might be added as a feature in future work by integrating geospatial and economic

indicators for existing and future transport options including road networks (Mosnier et al., 2014). Land and surface water resource availability is also added as an exogenous inflow into the system that must be continuously balanced by technologies and processes included in the model. This supports accounting for conservation measures that preserve land and move water downstream (environmental flows).

## 2.4 Enhancements to the MESSAGEix model

The existing MESSAGEix core model does not represent sub-annual storage dynamics and associated capacity constraints. Previous work demonstrates specific approaches for integrating short-term (i.e., daily) storage dynamics into long-term energy system models similar to MESSAGEix (Johnson et al., 2017); yet, sequential seasonal storage dynamics are most critical to represent from the perspective of water resources management, because of the important role reservoirs play in balancing seasonal hydrologic and demand variability, and the potential for future reservoir development to compete with other water uses

during filling. To enable inclusion of seasonal reservoirs in NEST, sequential monthly sub-annual time steps are included in the MESSAGEix implementation and the core model is enhanced with the following set of equations merged into the existing technical coefficient matrix and right-hand constraints vector:

$$\Delta S_{n,c,l,y,m} \cdot \Delta t_m + S_{n,c,l,y,m+1} - (S_{n,c,l,y,m} \cdot \lambda_{n,c,l,y,m}) = 0$$

$$S^-_{n,c,l,y,m} \leq S_{n,c,l,y,m} \leq S^+_{n,c,l,y,m}$$

$$S_{n,c,l,y,m} \leq Z_{n,c,l,y}$$

$$\Delta S_{n,c,l,y,m} \leq \Delta Z_{n,c,l,y} \tag{5}$$

In the above equations, $n$ is the node where the storage is located, $c$ is the commodity stored, $l$ is the level in the supply-chain

the storage interacts with, $y$ is the investment period (annual), and $m$ is the operational periods (sub-annual). The storage level is given by $S$, whereas the change in storage is given by $\Delta S$. The first set of inequality constraints is used to limit the storage level to within a specific range ($S^-$ is the lower bound and $S^+$ the upper bound), for example to include operating rules for reservoirs used for multiple purposes. The second and third inequality constraints are the capacity limitations both in terms of system size ($Z$) and rate of commodity transfer ($\Delta Z$). Storage losses (i.e. evaporation and seepage) are given by

the factor $\lambda$, and computed as a function of the estimated evaporation from the hydrological model and a linear area-volume relationship (Liu et al., 2018c). The sub-annual time period duration $\Delta t$ converts the storage change calculated as a rate into a volume consistent with the storage level. To account for filling behavior and interannual variations we ensure: (1) the start and end levels are the same across years when no new storage capacity is added; and (2) when new storage capacity is added, it must be filled uniformly throughout the first 10 years, thus presenting an additional freshwater demand. Capacity additions are

exogenously defined based on reported data; future work will consider the capacity limitations as control variables that can be expanded through increased investment in storage capacity.

To avoid integer (binary) decision variables associated with the choice of whether or not to plant a specific crop in a specific area, an additional set of minimum utilization constraints are defined for crops included in MESSAGEix. This forces the optimization to maintain the growing schedule over the course of the year, while balancing the total land area across crop types. Further adjustments to the core model are needed to ensure the physical balance of EWL resources. Specifically, the existing MESSAGEix core model constrains resource supply to be greater than or equal to resource demand. This setup enables the model to spill excess resource production when beneficial to the overall operating costs of the system. However, this configuration poses challenges when accounting for inflows into the system to effectively size infrastructure capacity. For example, when considering wastewater return flows as a specific commodity that should be managed using wastewater treatment technologies, it is crucial to ensure a complete commodity balance across all time periods. Otherwise, the model would be able to exclude inflows to avoid building wastewater treatment capacity. To reconcile inconsistencies and to ensure a physical balance of EWL resources, we define a new set of supply-demand balance equality constraints in the enhanced MESSAGEix core model used in NEST.

Finally, for computational efficiency we developed a set of tools in the R programming interface that enable users to rapidly prototype new models during the testing phase by selectively managing interactions with ixmp. We found that for the case study described in Section 3 that the new approach cuts model instance generation time by an order of magnitude. Importantly, the ixmp utilities can be optionally used so that once debugging is complete, models can readily be shared and modified using the powerful database utilities enabled with ixmp. All of the enhancements to the MESSAGEix model implemented in this paper can be obtained from the online repository for NEST (https://github.com/iiasa/NEST).

## 3 Modeling SDG implementation in the Indus River Basin

As a first application of NEST, we focus on the Indus River Basin (IRB). The setup is meant to demonstrate the capabilities of the model, with ongoing work dedicated to the integration of local data and understanding of the policy implications for the region, and to be summarized in a future publication. The IRB, located in South Asia, is home to an estimated 250-million people (Pakistan 61%, India 35%, Afghanistan 4%, and China less than 1%) and has the highest density of irrigated land in the world (Laghari et al., 2012; Yu et al., 2013). In recent years, the region experienced rapid population and economic activity growth, and this is expected to continue in the next decades leading to reduced poverty and growing demands for water, energy and food. With no surface water left in the basin for expanded use and accelerating exploitation of fossil groundwater as a result, long-term management of systems dependent on water is fundamental for the sustainable development of the region (Wada et al., 2019).

There have been a number of previous analyses of EWL challenges in the IRB, including integrated modeling of the systems in Pakistan's portion of the basin to understand the cost of climate change (Yu et al., 2013; Yang et al., 2016). Other recent analysis has quantified existing and future gaps in water supply caused by projected socioeconomic and climate change or

gaps in estimating electricity demand variation due to groundwater pumping for agriculture (Wijngaard et al., 2018; Siddiqi and Wescoat, 2013). Previous work on the IRB does not provide a full assessment of EWL adaptation options or long-term pathways for the IRB as a whole. Specifically, there remains a need to link long-term capacity expansion decisions across EWL systems to understand the best strategies for developing the region's infrastructure into the future while accounting

for existing transboundary policies. Crucially, there are important interplays between irrigation efficiency, land-use change and groundwater recharge that need to be reconciled to ensure water saving policies have the intended effect (Grafton et al., 2018). The NEST framework is ideally positioned to tackle these research questions because of its explicit representation of EWL capacity expansion and land-use change across spatially distributed regions and features basin wide water accounting for surface and groundwater systems.

## 3.1  Model setup

To parameterize the model in terms of resources, technologies and demands, we used the data sources outlined in Table 1. Importantly, much of the data needed to run NEST can be obtained from open access geospatial datasets with global coverage. Thus, NEST is readily adapted to other regions of the world. Nevertheless, it is important to emphasize the prioritization of approved local data, as well as use of the calibration steps that can be embedded in the framework that improve the performance

of the model in terms of reproducing historical conditions. Moreover, it is important to stress the use of multiple climate models and RCP-SSP scenarios to bridge the range of uncertainties in the hydrological modeling and demand drivers.

We calibrated CWatM for the IRB at 5 arcmin resolution using the monthly streamflow data during 1995-2010 at the Besham station, in northern Pakistan.The Besham station is chosen because of its coverage of historical years, it incorporates the runoff from both glacial and seasonal snowmelt. However, multiple stations would be necessary to better represent regional hetero-

geneity (in particular lower versus upper basin). Future work will incorporate spatially distributed observations to improve the calibration. It is important to emphasize the complexity of the hydrology in the IRB and the difficulties in calibrating to observed data due to extreme elevation changes (Forsythe et al., 2019). For calibration, the CWatM simulations included human impacts on streamflow and a spin-up period of 5 years to allow long-term storage components to stabilize. Analysis of the initial calibration results showed that the calibration was mainly impacted by the ice melt coefficient and empirical shape parameter

of the ARNO model for infiltration (Todini, 1996; Burek et al., 2013). Therefore, we ran a second calibration that searched for optimal values for only these two parameters. The calibrated parameter values are given in Table 2. The performance of the model after the two calibration runs is in Figure 6. We then used the calibrated CWatM for the IRB for historical (1956-2005) and future (2006-2099) simulations using the downscaled meteorological inputs of the ISI-MIP2b project from four global climate models (GCMs: GFDL-ESM2M, HadGEM2-ES, IPSL-CM5A-LR, MIROC5) (Frieler et al., 2017). The streamflow in

CWatM were naturalized because human activity, and water withdrawals in particular, are represented and accounted for in the MESSAGEix framework. The resulting ensemble mean monthly runoff profiles for each riparian country's basin area and the Indus as a whole are depicted in Figure 7. The total basin runoff matches closely with other reported data (Laghari et al., 2012).

For implementation in MESSAGEix, the IRB is delineated into 24 Basin Country Units (BCUs) using the basin and country administrative boundary datasets (Figure 2). Further disaggregation into the agro-ecological zones is not pursued in this case

| Parameter(s) | Dataset | Spatial Resolution | Latest Year |
|---|---|---|---|
| Country administrative boundaries | Database of Global Administrative Areas (GADM) | polygon | 2008 |
| Basin and sub-basin boundaries | HydroBASINS database | polygon | 2012 |
| Climate forcing | Intersectoral Impact Model Intercomparison Project (ISIMIP) | $0.5° \times 0.5°$ | 2015 |
| Urban and rural population | Jones and O'Neill (2016) | $0.125° \times 0.125°$ | 2010 |
| Urban and rural GDP | Byers et al. (2018) | $0.125° \times 0.125°$ | 2010 |
| Elevation, flow-direction, basin/lake boundaries | HydroSHEDS Database | $0.004° \times 0.004°$ | 2008 |
| Non-hydro power plant capacity, age and location | World Electric Power Plant (WEPP) Database | asset-level | 2017 |
| Power plant cooling technologies | Raptis et al. (2016) | asset-level | 2014 |
| Hydro power plant capacity, age and location | van Vliet et al. (2016) | asset-level | 2017 |
| Reservoir capacity, age and location | Global Reservoir and Dam (GRanD) Database | asset-level | 2014 |
| Crop areas, yields and location | Global Agro Ecological Zones (GAEZ) Database | $0.1° \times 0.1°$ | 2005 |
| Protected areas | World Database on Protected Areas (WDPA) | polygon | 2014 |
| Forests | Global Forest Change (GFC) Database | $0.004° \times 0.004°$ | 2014 |
| Depth to groundwater | Fan et al. (2013) | $0.01° \times 0.01°$ | 2012 |
| Historical energy supply and demand by sector | International Energy Agency (IEA) | national | 2017 |
| Historical water supply and demand by sector | Information System on Water and Agriculture (AQUASTAT) | national | 2015 |
| Historical irrigation water supply by source | Cheema et al. (2014) | $0.1° \times 0.1°$ | 2015 |
| Historical non-irrigation groundwater use | Wada et al. (2016) | $0.5° \times 0.5°$ | 2005 |
| Historical transmission capacity and roads | OpenStreetMap | asset-level | 2017 |
| Historical on-farm energy use incl. pumping | Siddiqi and Wescot (2013); Rao et al. (2019) | provincial | 2015 |
| Historical water conveyance capacity | Estimated from technical reports | asset-level | 2018 |
| Historical crop prices, fertilizers and crop coefficients | Food and Agriculture Organization (FAO) | national | 2018 |
| Planned reservoir and power plant capacity | Estimated from technical reports | asset-level | 2030 |
| Power plant cost and performance | Parkinson et al. (2016); Fricko et al. (2016) | technology-level | 2014 |
| Surface and groundwater performance | Kahil et al. (2018) | technology-level | 2010 |
| Irrigation cost and performance | Local data collected | technology-level | 2010 |
| Wastewater cost and performance | Parkinson et al. (2016) | technology-level | 2014 |
| Desalination cost and performance | Parkinson et al. (2016) | technology-level | 2014 |

**Table 1.** Data sources leveraged to parameterize the NEST implementation of the IRB.

because of limited spatial variability in crop potential within the delineated BCUs. The planning horizon considers investment periods spanning 2020 to 2060 in 10 year time steps, and 2015 is parameterized as the base historical year (i.e., the initial starting point). Monthly sub-annual time steps are considered.

CWaTM is run with fixed spatial and temporal resolution as mentioned in previous sections. Therefore, performances are not

5    affected by the final scale of the optimization model. Running times are in the order of few hours on personal computers. The MESSAGEix component is instead scale sensitive, increasing the number of BCU or the temporal resolution increases the size of the matrix of the LP optimization significantly. In the current configuration, the *cplex* solver in the GAMS model reduces the system of equations to a LP matrix of approximately 1 million x 1 million elements and solves in less than 30 minutes on

| Parameter | Value |
| --- | --- |
| Snow melt coefficient | 0.003597 |
| Crop factor correction | 1.211 |
| Ice melt coefficient | 0.5366 |
| Soil preferential flow constant | 5.4 |
| ARNO b | 1.259 |
| Interflow part of recharge factor | 1.807 |
| Groundwater recession coefficient factor | 3.823 |
| Runoff concentration factor | 1.492 |
| Routing Manning's N | 8.104 |
| Reservoir normal storage limit | 0.5257 |
| Lake alpha factor | 1.154 |
| Lake wind factor | 1.205 |

**Table 2.** Calibration parameters values for convergence of CWatM.

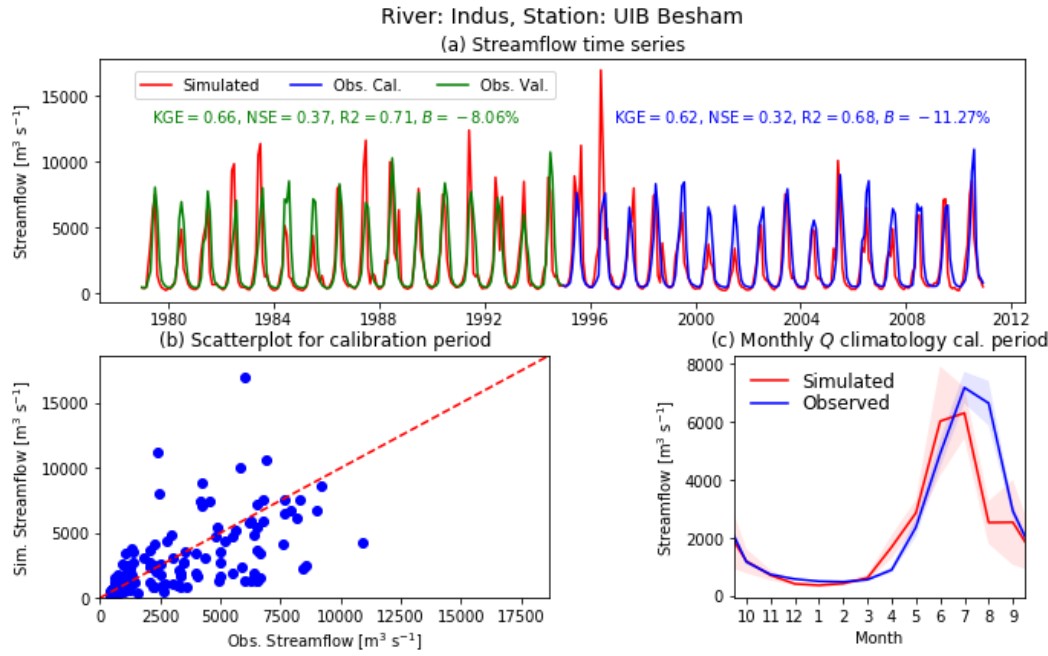

**Figure 6.** Comparison between the simulated streamflow by the calibrated model and the observation. KGE: Kling-Gupta Efficiency. NSE: Nash-Sutcliffe Efficiency. R-sq: R-square. B: mean bias.

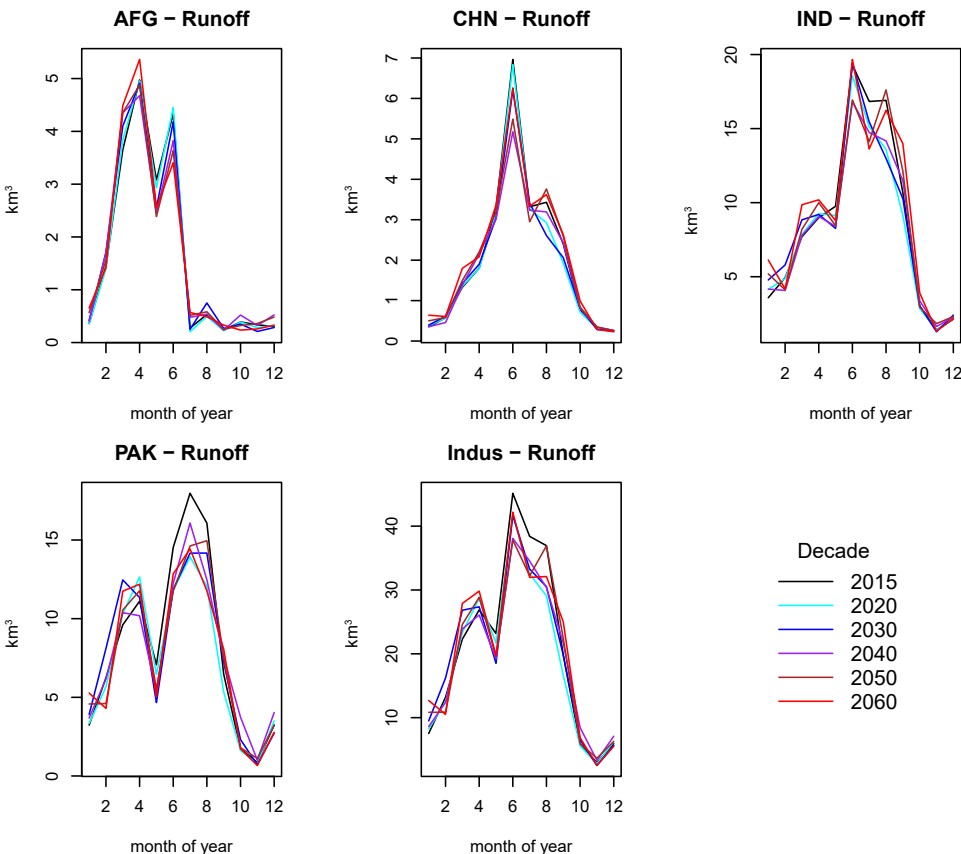

**Figure 7.** Ensemble mean monthly runoff in each country and the IRB as a whole. Daily run-off sequences from CWatM are converted to decadal runoff scenarios by averaging monthly gridded volumes over a 30-year time period. Outputs from four global climate models are included in the ensemble: GFDL-ESM2M, HadGEM2-ES, IPSL-CM5A-LR, MIROC5.

personal computers. For each policy scenario described in the following sections, CWaTM is only run once for each SSP and RCP combination, while additional policies are only implemented and run in the optimization model.

With most of the land area dedicated to crop production, we simplify the reference system by limiting the land-use options to crop land choices and limit the crop types to fertilized options. The SSP2 (middle-of-the-road) socioeconomic scenario is
5  explored in the analysis and the *ensemble* mean climate scenario across the RCP climate models is used for climate forcing.

Urban and rural population and per capita income for SSP1, 2 and 5 projected for 2050 are compared to 2010 values for each riparian country's part of the IRB in Figure 8. It can be seen that rapid urbanization and growth in income levels is projected in the scenarios, and these changes translate into increased consumption of water, energy and crops in the modeling framework. Figure S1 depicts the corresponding sectoral exogenous demands for the SSP2 scenario. Note that results for China
10 are not included because the existing and projected population growth in this region is very low and thus the consumption has

negligible impact on the downstream resources. Electricity demands increase most dramatically across countries due to the rapid increases in GDP and the assumption that electrification is supporting economic development. Water demands increase more gradually due to less influence of economic growth, although for India the manufacturing sector water uses increases significantly due to the existing water intensity. Corresponding projections of the population with and without access to pre- and post-treatment of freshwater are generated based on the GDP projections.

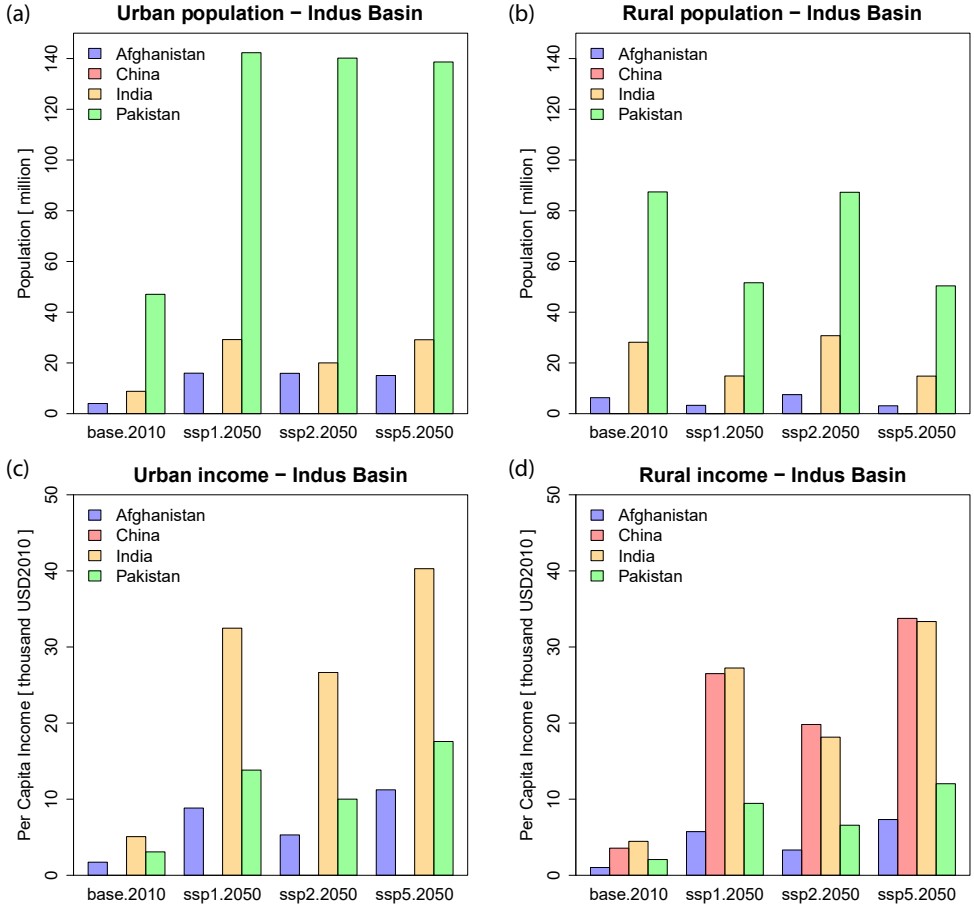

**Figure 8.** Urban and rural population (a),(b) and per capita income (c), (d) for SSP1, 2 and 5 in 2050 and 2010 for each riparian country's part of the IRB.

Canals play an important role in enabling the Indus Water Treaty, and are mapped to specific BCUs using the data in Table S1. Operational constraints are also added to force the linkages to transfer water between routes, in line with the Indus Treaty. The Indira Gandhi canal is considered as a constraint on flows originating from the particular BCU where the inlet is found. Similarly, an urban water transfer to Karachi near the Indus Delta is included as an additional demand. The capacities of other

water diversion infrastructures (surface and groundwater) for each sector are estimated from the historical withdrawals. The energy source for groundwater pumping is also identified, where diesel generators dominate in Pakistan and Afghanistan, and electricity is used predominately in India.

The existing and planned capacity of power generation in the IRB is depicted Figure S2. Hydropower is the main source of generation capacity in the basin, with the basin regions of Pakistan also hosting significant amount of fossil generation. A number of large-scale hydropower projects are also planned in the region (Table S2) that were not found in the global databases, and these projects have also been included in the model. For projects with an opening date before 2025, it is assumed they are operational in 2020; all other projects are assumed to be operational in 2030. For hydropower projects with storage, the storage capacity is added to the BCU level storage in the year it becomes operational, with the filling of the reservoir averaged over the first 10 years of operation, as described previously. Existing storage capacity includes 26.4 km$^3$ in Pakistan, 22.2 km$^3$ in India and 0.6 km$^3$ in Afghanistan. Operating rules are derived for the largest existing dams based on the historical reported releases for 2016 and 2017. Approximately 45 GW of additional hydropower potential is estimated using the approach described in Section 2.3.2, mainly in the Upper Indus Basin. The assessed solar and wind potential greatly exceeds the electricity demand, with most of the wind potential focused mainly in the Indus Delta region. Tapping the solar and wind potential, however, requires investment in transmission and flexible assets (e.g., storage). In the Supplementary Information we provide variable capacity factor of solar and wind aggregated for each BCU (*Variable_capacity_factor.xlsx*).

The performance of the different crop types considered in the model in terms of yields are presented in Figure S3 for irrigated and non-irrigated options. Crop categories are set according to the main types of crops grown in the region, with some aggregation of crop types occurring to simplify the number of decision variables. The maximum productivity on a per hectare basis demonstrates that irrigation significantly boosts crop productivity in many locations, enabling less land to be used. As mentioned previously, land for each crop type in each BCU is constrained based on suitability and total area. Certain crops also are performing better than others in some regions, while some crops are not available entirely in some regions. The historical crop yields are harmonized to historical irrigation water use by calculating the required irrigation to support the historical crop areas using Eq. 3, and then calibrating the irrigation intensities such that the withdrawals match with the reported irrigation deliveries in Cheema et al. (2014) aggregated to the BCU-scale.

## 3.2  Scenario analysis

The parameterized NEST model of the IRB is applied within a scenario analysis in which a baseline (business as usual) scenario and a multi-objective scenario achieving multiple SDG indicators by 2030 are compared. The SSP2 information is used to parameterize population and economic indicators in each scenario. The business as usual scenario assumes the continuation of existing policies (e.g., Indus Water Treaty), and is aiming at cost minimization with limited environmental constraints such as emission or infrastructure access targets. Conversely, the SDG implementation pursues a vision of economic growth (poverty eradication) jointly combined with reducing resource access inequalities and the environmental impacts of infrastructure systems. It is important to emphasize the SDG scenario is not exploring all of the individual targets and indicators, but instead a limited set relevant for water, energy and land systems that are also well represented in the NEST framework.

The main features of the baseline and multiple-SDG scenarios are summarized in Table 3. The scenarios are simulated by solving NEST under the different implementations. Additional sensitivity analysis is performed to highlight uncertainties in the modeling framework.

| Target | Description | Modeling: SDG vs baseline |
|---|---|---|
| Climate action | | |
| Global Greenhouse gas (GHG) Emissions | SDG 13.a Implement the commitment undertaken by to the United Nations Framework Convention on Climate Change | Set GHG emission budget and climate scenario accordingly. Baseline: no emission targets |
| Clean and affordable energy development scenarios | | |
| Clean energy access | SDG 7.2 By 2030 50% share of renewable energy in the global energy mix | Set targets on share of renewables (wind, solar, geothermal). Baseline: no targets |
| Power plant cooling | SDG 7.b By 2030, expand infrastructure and upgrade technology for supplying modern and sustainable energy services for all | Phase out of once-through cooling, imposing capacity constraint. Baseline: no targets |
| Water sector development scenarios | | |
| Sustainable water withdrawals | SDG 6.6 By 2020, protect and restore water-related ecosystems, including mountains, forests, wetlands, rivers, aquifers and lakes | Minimum of 20% of monthly natural flow left in rivers and aquifers by 2030. Set sustainable levels of groundwater extractions (also in baseline) |
| Wastewater treatment | SDG 6.3 By 2030, improve water quality by reducing pollution, halving the proportion of untreated wastewater and substantially increasing recycling and safe reuse globally | Treat half of return flows treated by 2030, recycle one quarter of return flows. Baseline: no targets |
| Sustainable agriculture scenarios | | |
| Food & agriculture infrastructure access | SDG 2.4 By 2030, 100% implementation of modern so-called smart irrigation technologies that increase productivity and production relative to 2015 | SDG 2.4 constraint technologies with low efficiency to have zero capacity in 2030. Baseline: no smart irrigation technologies adopted before 2030 |

**Table 3.** Policy scenarios embedding specific SDG targets, (SDG)

## 4  Results and discussion

### 4.1  Role of water storage and seasonal effects

Figure 9 shows the balance of surface water in the model for a specific sub-catchment (BCU) in Pakistan with planned storage expansion in 2030. Inflows in the region from upstream river flows or from internal runoff are subject to strong seasonal variations (a). Urban, rural and industry water demands are assumed to be constant through the year (therefore not shown in Figure 9), while water requirements for agriculture and power plants' cooling are instead endogenous and thus variable during the year (c). Supply and demand are not constrained to specific water sources, Figure 9 depicts a case in which water

requirements for agriculture are supplied by groundwater (b). Most surface water is indeed outflowing from the region to downstream nodes (d) due to environmental flow requirement and it is in turn used for hydroelectric generation (e). Noticeably, storage absorbs the high inflow peaks in the months of April and June, and releases high outflows in July (f). However, it is not straightforward to directly link the reservoir level changes to hydropower generation or other regional water requirements

5 under the conjunctive management strategy. Storage regulation appears in this case to mostly be serving downstream water demands as opposed to supporting hydropower potential.

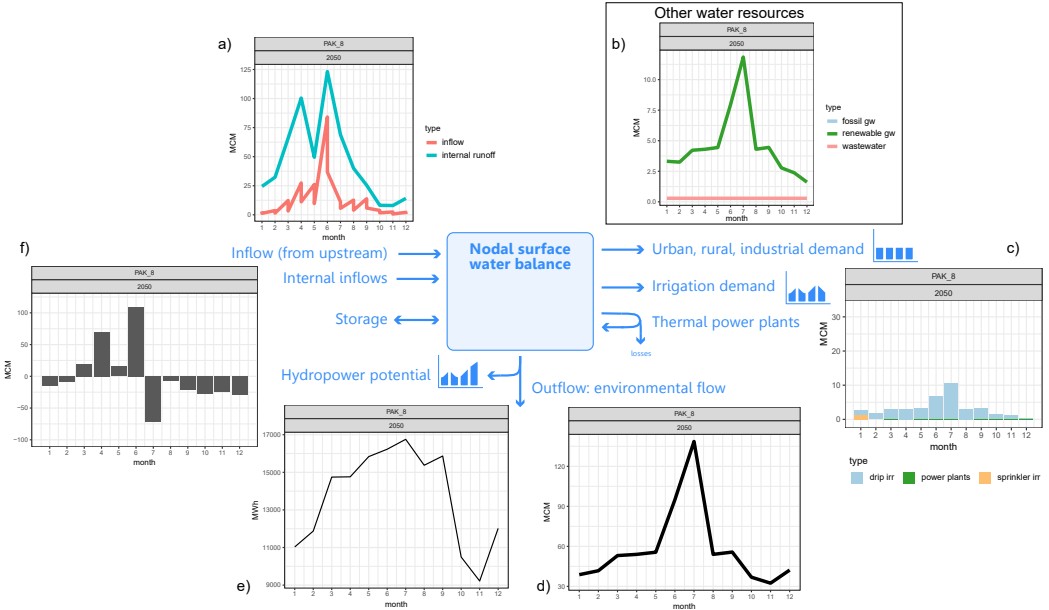

**Figure 9.** Surface water balance for a specific BCU (PAK_8) in 2050 (a) surface water inflows (b) supply from other water sources (c) variable water demand (d) surface water outflow (e) hydroelectric generation (f) storage level changes, recharging if positive, discharging if negative.

Seasonality effects embedded in the model input are mostly related to water availability, renewable energy capacity factors and crop water requirements and productivity. Figure 10 shows outputs of the model that are affected by the above mentioned seasonal variations. Electricity generation fluctuations in hydropower generation are mostly compensated by nuclear, imports

10 or natural gas. Similarly, the time for crop cultivation, growth and yield is season specific, taking into account precipitation and crop coefficients seasonality. Other studies have looked at the role of hydropower in the region with a nexus perspective, considering both electricity production and water management (Yang et al., 2016). Whilst the results from the Indus Basin Model Revised (IBMR) and NEST could be compared, if similar scenarios were run, it must be noted that IBMR only focuses on a sub-region of the basin network with higher spatial detail, while NEST includes a more complete representation of energy

15 demands, supply and water-energy linkages.

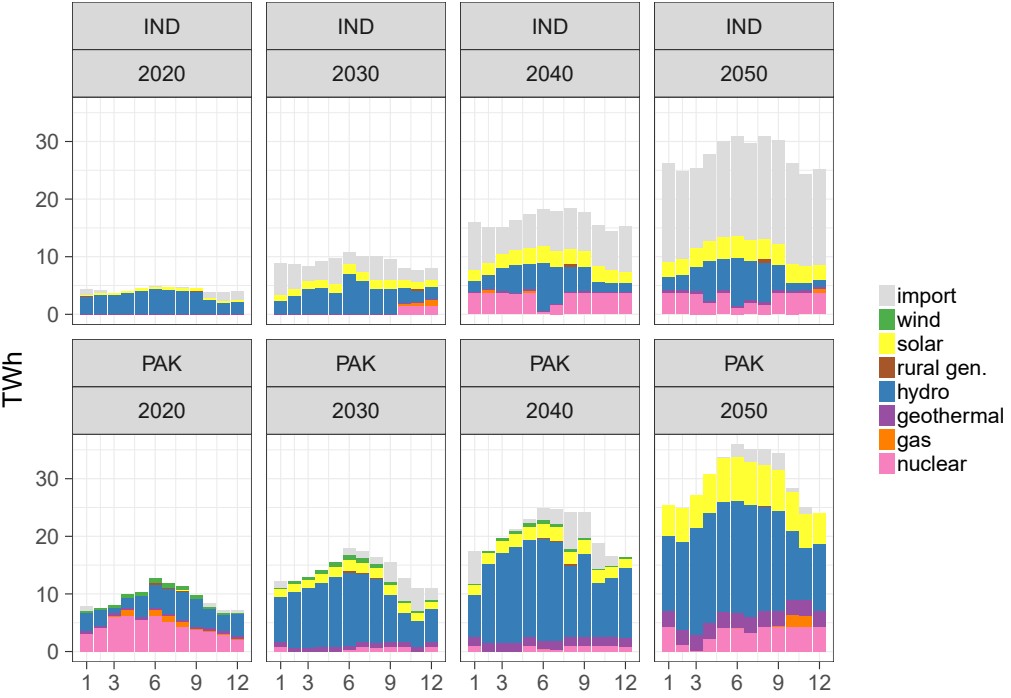

**Figure 10.** Monthly electricity generation under the *multiple SDG* scenario

## 4.2 Quantifying investments to achieve the SDGs

We present a comparison between the baseline and the multiple SDG scenarios. Figure 11 (a) depicts the yearly average new investment portfolio and associated average operational costs for each scenario. To achieve the sustainability goals, investment costs approximately double, while operational costs increase by about 30% (these include fixed costs, variable costs of opera-
5   tion and costs of electricity imports). To meet the targets for wastewater treatment and the share of renewable energy (solar and wind and geothermal), a large portion of the new investments are dedicated to technology development and a shift in power plant type. Hydropower emerges as an important option in the baseline because of the planned expansions and unexploited potential quantified in the assessment.

As a consequence of the environmental flow policy in SDG6, multiple sectors need to adapt to lower water availability.
10  The agriculture sector is particularly impacted due to its high share of total water demands and expands to non irrigated areas to avoid water withdrawals. However, this implies lower yields and so more area is needed to support the same production and at higher operational costs due to the lower productivities. Additionally, there is increased investment into more efficient irrigation technologies, especially where most of the available arable land is cultivated and production still needs to be boosted to maintain agricultural supplies.

Figure 11 (b) compares the nexus interactions at the basin-scale for each scenario. The multiple SDG scenario displays an almost fivefold increase (from 500 GWh to 2500 GWh per year) in energy requirements for water management (mostly pumping, treatment and arrangement of new canals). This is to support increased water access in the municipal sector and massively expanded wastewater treatment capabilities in urban areas, but still represent less than 2% of total electricity generation projected in 2020. A combined GHG emission target ensures the increased demands are met without increasing carbon emissions.

These results demonstrate the value of interconnection across EWL sectors in terms of chain reaction in investments (i.e. expanding piping distribution also require expansion in electricity production and distribution), synergies (investing in irrigation efficiency implies saving in water distribution for irrigation) and trade-offs, as it is clearly not possible to minimize costs and resource use across all sectors to achieve the SDGs.

## 4.3 Synergies and trade-off among SDG targets

The sustainability scenario includes multiple policy objectives across different sectors, which are considered simultaneously by the model. Specific policy objectives can thus be analyzed individually or in combination. Cross-sectoral implications are not necessarily the same when assessing multiple policies at the same time or individually. However, to additionally understand the implication of each single SDG policy on the water, energy and land systems, we tested each policy independently (as in Table 3).

Figure 12 depicts the electricity generation, water withdrawal by source and the land use for agriculture in India and Pakistan from 2020 to 2050 in all the scenario permutations tested. The baseline scenario assumes that enough water is present in the basin to meet increasing energy, water and food demands, while fulfilling the Indus Water Treaty allocations, but neglecting the additional environmental flow standards, water efficiency guidelines and infrastructure access constraints present in the SDG6 case. The second row of plots depicts the sectoral changes induced by the multiple sustainability policies. Intuitively, constraining the use of surface water for environmental purposes has most impact on cross-sectoral activities in Pakistan because it is the most downstream country and thus faces the greatest challenge in meeting increasing water demands while concurrently allocating more flow to ecosystems when water is already scarce. In fact, its hydroelectric potential is significantly reduced and the main water source left is renewable groundwater. This has a large impact on the agriculture system, where both India and Pakistan expand cultivated land with rain-fed crops, to adapt to water scarcity[1].

It is crucial to note that in India the total available land for agriculture is already utilized in the base year in most of the modeled regions due in part to the Indus Water Treaty obligations (which allows India to use a limited amount of western river waters for irrigation). Thus, to fulfill increasing food demand and reduce the water consumption per hectare in the SDG scenario, an uptake in more efficient irrigation technologies is observed. Importantly, the basin-wide water accounting framework enables the applied water efficiency policies to account for the complex interactions between irrigation water losses and groundwater availability, to ensure that a combination of surface and non-renewable groundwater sources are conserved.

---

[1]For this case study we do not consider land use change to other types of land, such as forests

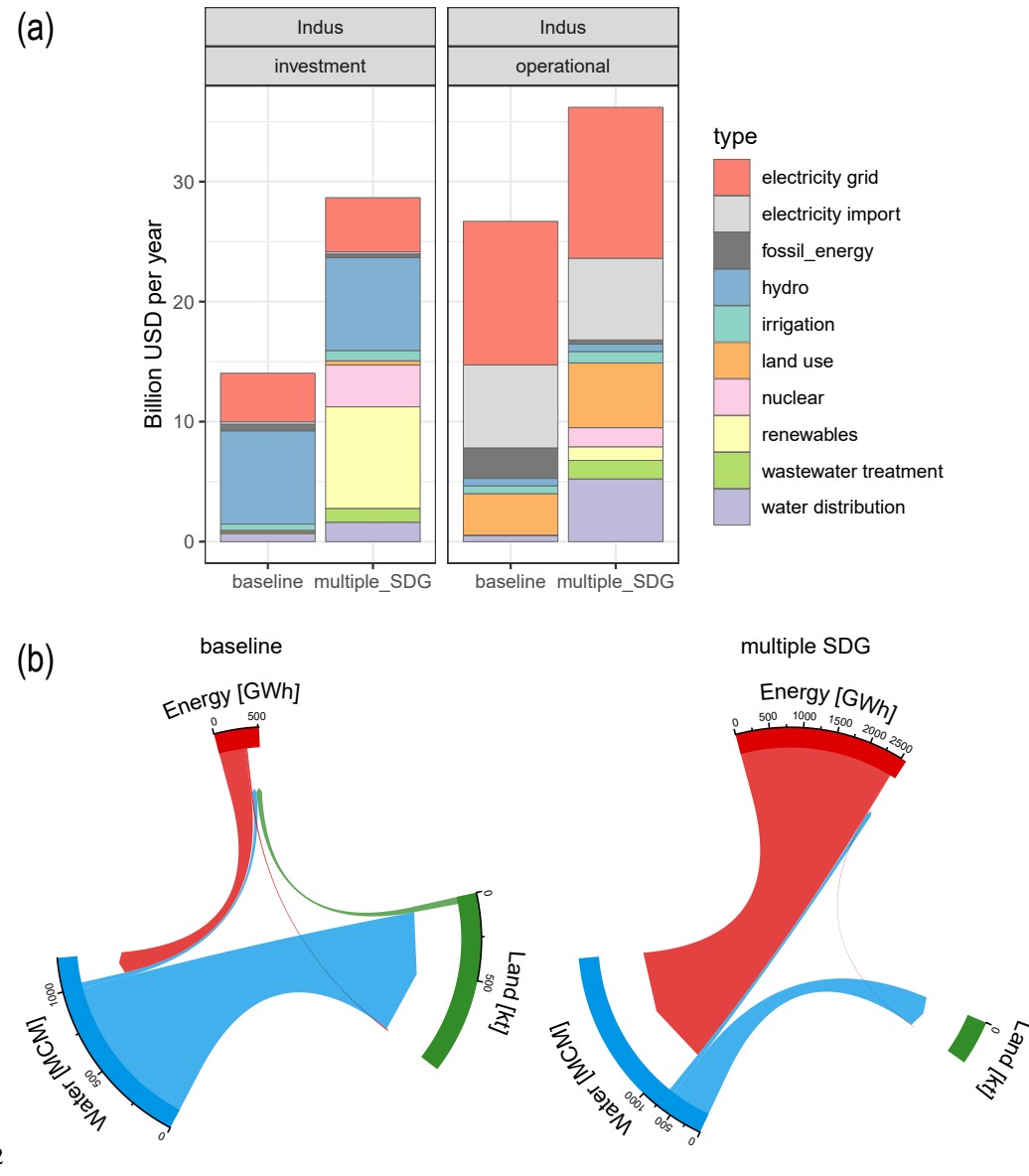

**Figure 11.** (a) Comparison of investment and operational cost (yearly average) for the entire basin. (b) Representation of water for energy (excluding hydropower) and irrigation [MCM], energy for water management technologies and for agriculture [GWh] and biomass used for energy production [kt].

Looking at single scenarios separately helps to understand what policy drives the specific changes and what sector is mostly affected.

- SDG2. Most of the existing flood irrigation systems are substituted by drip and sprinkler technologies. This reduces the water demand for irrigation. For further analysis the authors intend to add other SDG2 related targets concerning changes in food demand, import, export and shifts to different types of crops.

- SDG6. The environmental flow policy represents one of the major constraints for the resource management in the region. Indeed, we notice how, particularly in Pakistan, electricity and water supply systems would require complete restructuring, as well as management of land for agriculture. The main water resource for Pakistan becomes renewable groundwater, which is recharged from via infiltration including losses from irrigated fields. One important difference to the multiple SDG scenario is the role of hydropower and the consequences on the remaining surface water availability in Pakistan. In fact, as the SDG6 scenario is not bound by emission constraints, fossil fuel generation (gas and oil) is rapidly deployed. When adding $CO_2$ emission and renewable energy shares consistent with SDG7, results show it can be optimal for Pakistan to exploit all the possible hydropower potential, while meeting environmental flow minimum requirements. This reduces the surface water availability both for irrigation and other demands. As a consequence, less irrigation technologies are adopted in the multiple SDG scenario in favor of more rain-fed crops. However, this leads to a vicious circle where less irrigated land means less water recharging groundwater aquifers, but at the same time the model accounts for the interaction and finds an optimal balance.

- SDG7. This policy imposes specific targets for solar, wind and geothermal electricity production in terms of the share in the entire energy mix. We set the share target of 30% by 2050, which is achieved gradually starting with 10% in 2020. In addition, a phase out of coal and once through cooling technologies after 2030 are also considered. One consequence of this policy is a more rapid transformation away from fossil fuels. Nonetheless, this is not necessarily the most economically optimal way of achieving $CO_2$ emission reduction (see SDG13). When compared to the multiple SDG scenario, nuclear plays a more significant role, despite higher water consumption. Since nuclear is currently a critical issue in both India and Pakistan, further research will investigate the feasibility of nuclear with more detail and interacting with local stakeholders.

- SDG13. To understand what are the possible pathways towards a carbon neutral electricity system, the SDG13 results show how nuclear electricity generation can be an important option due to cost and reliability, and is complemented well by the available hydropower potential. Importantly cost and policy barriers difficult to monetize in the framework could cause development constraints for nuclear systems in the region.

In summary, this overview of the single policy objectives shows that constraints on land and water availability push the system to make transformational changes to the development pathway for each sector, and can drastically alter the structure of the energy and water supplies and land-use pattern. Considering multiple target simultaneously shows different results than summing individual analysis. As mentioned above, the electricity mix changes when considering water constraints and climate targets.

Similarly, land use is different when efficiency policies are in place together with environmental targets. This clearly shows the importance of an integrated multi-sectoral analysis to highlight synergies and barriers among objectives. The authors intend expand this topic in upcoming research.

## 4.4 Uncertainty and sensitivity

Integrated assessment models are subjected to different types of uncertainty, which can cumulate and therefore require particular attention. Uncertainty can be broadly divided in data or parametric uncertainty, which is given by data sources, often represented as distribution or numerical ranges; and assumption uncertainty, occurring when dealing with future scenario in the scope of policy analysis (Rotmans and van Asselt, 2001).

Here we present an example of scenario uncertainty propagation between the two different models in NEST and a simplified parametric sensitivity analysis to demonstrate the need for a more thorough study.

Figure 13 (a) shows different level of monthly total runoff from the CWaTM using different climate models and under two different climate scenarios (RCP 2.6 and 6.0). We notice major diversity in trend given by different climate models, while climate scenario implies changes mostly in the eighth and ninth months of year 2020. When running the optimization model in NEST, outcomes carry the uncertainty from the hydrological model and cumulate it with other types of uncertainty. Figure 13 (b) shows total cost for the Indus region where the uncertainty of different SSP assumptions is added the previous set of climate scenarios. We notice how SSP assumptions more greatly affect total cost compared to either climate model or RCP (each bundle of same-color lines includes runs with all climate scenario and RCP assumptions). However, looking at SSP 2 and 1, with reduced stress caused by population growth, climate uncertainty is more significant than for SSP 5.

Figure 14 illustrates different output changes under the BAU scenario in response to an arbitrary variation in input parameters (-25%,+25%). Intuitively, some outputs as groundwater extraction and energy production are strongly affected by the variation of sector-related parameters, irrigation and power plants' efficiency respectively. The plot also shows some significant cross-sectoral feedbacks. For instance, changes in energy efficiency and investment cost impact groundwater and surface water withdrawals by up to 5% and irrigation efficiency strongly impact fossil-fueled energy production. Land use seems not to be sensitive to the input parameters here considered. Looking at total cost alterations, multiple parameters induce variations comparable with the scenario uncertainty described above. This preliminary analysis suggests that further and more thorough study is needed, adopting more realistic uncertainty ranges or data distributions.

Structural uncertainty also typically characterizes complex models such NEST (Ajami et al., 2007). We expect that omitting some feedbacks or expanding some modules could distort some of the model responses. Further work will focus on exploring this type of uncertainty.

## 4.5 Stuctural limitations and further developments

Increasing spatial and temporal resolution might be helpful to focus on sub-regions and identify possible critical areas with higher detail. However, it brings greater computational challenges associated with using classical mathematical programming methods. In this context, scaling of the input-output coefficients to ensure fast solution times can be challenging for nexus

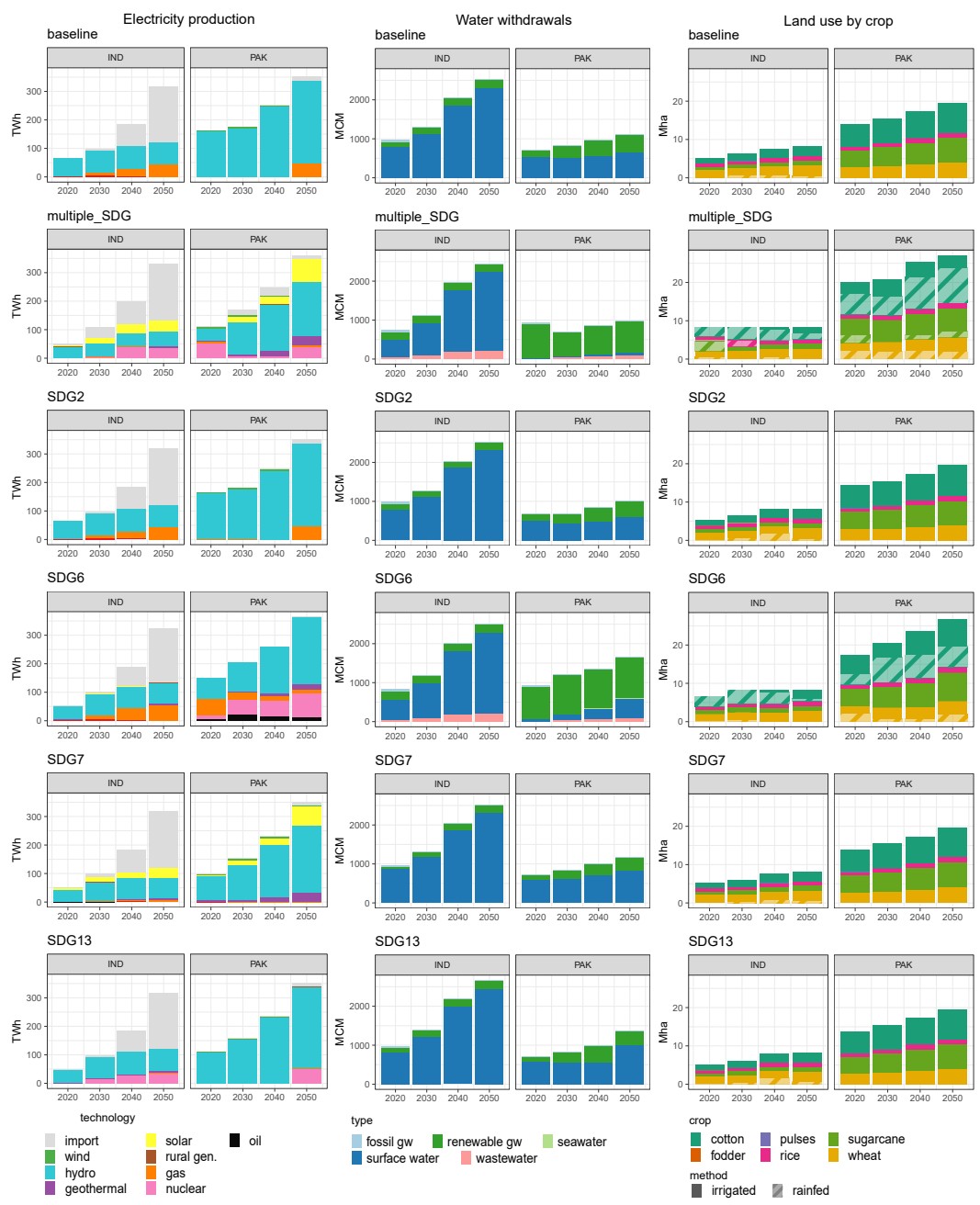

**Figure 12.** Comparison among different scenarios of yearly values for Pakistan (PAK) and part of India in the Indus basin (IND) of: electricity supply mix [TWh], water withdrawals from different sources [MCM], total land used for farming different crops [Mha], distinguishing between irrigated (dark color) and non irrigated area (semi-transparent)

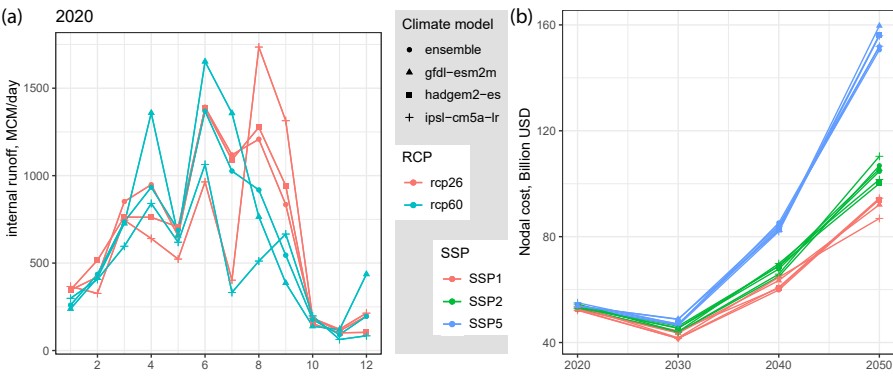

**Figure 13.** Total monthly runoff availability in 2020 under different RCP and climate models (a). Total costs under different climate model, RCP and SSP assumptions (b)

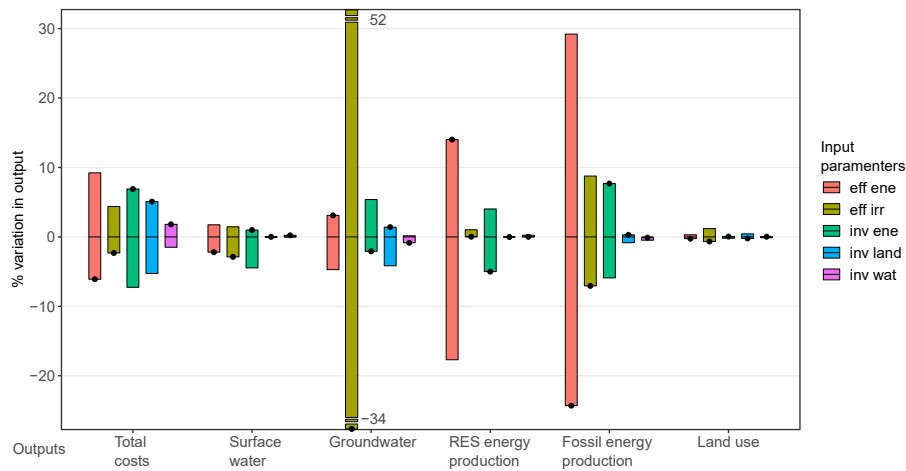

**Figure 14.** Changes in model outputs in response to a variation (-25%,+25%) of input parameters, namely power plants' efficiency (eff ene), irrigation technologies' efficiency (eff irr), investment costs of power plants and transmission lines (inv ene), irrigation and land management (inv land) and water distribution/treatment (inv wat). The outputs refer to average yearly values for basin's total system costs, surface and ground water withdrawals, energy production from renewable or non-renewable sources and land use for agriculture. The black dots show the outputs corresponding to +25% variation in inputs.

models, because many cross-sector interactions require definition of input-output coefficient ranges covering multiple orders of magnitude. Future work may need to explore heuristics or other emulations as an alternative approach to classical optimization methods in order to integrate and optimize the vast amounts of geospatial data increasingly available and promoting the use of ultra-high resolution models for infrastructure planning.

From a hydrological perspective, some limitations of the current NEST formulation include the use of static land-use maps in the development of the water resource potentials. Dynamic land-use maps could be used in future work using the optimal solutions from MESSAGEix. An important next step involves downscaling water- and land-use results to the spatial scale used in the hydrological model, improving the visualization and analysis of results, as well as enabling spatially explicit calculation of water availability and demands to represent dynamic changes of water and land-use consistently across the two models in NEST. The assessment of groundwater could also be improved by including lateral groundwater flows and by changing the representation of aquifer recharge to a non-linear model. A major constraint in modeling hydrological processes is the linear formulation of the optimization model which limits dynamic representation of key sustainability indicators as continuous model decision variables (e.g., water quality).

Finally, assumptions on boundary conditions, such as costs of imports (of food, electricity or water), are important for simplistic assumptions (e.g. electricity imports in Figure 12). Future work could improve the representation of boundary conditions with supply-cost curves or by linking with market models representative of the system outside the study area. Linking with global and regional integrated assessment models through the common commodity markets could improve the expected import-export response in NEST under scenarios of global change and explore different scenarios of basin self-sufficiency and resilience to external shocks.

## 5   Conclusions

The NExus Solution Tool (NEST) links a distributed hydrological model with a multi-sector infrastructure optimization model, the framework of which described in this paper in detail and applied to the Indus River Basin's energy, water and land systems. The framework is flexible and can be adapted to other regions of the world. NEST is designed to produce indicators relevant to the SDGs for water, energy, land and climate and to tap into the increasing volumes of geospatial data openly available through national inventories and the earth system modeling community. Comparing results for a business as usual scenario to one where multiple SDGs are enforced highlights the framework's capability to capture clear differences in the optimal investment portfolio and cross-sector interactions characteristic of the SDGs.

A key innovative feature of the NEST framework is the dynamic linking of the distributed hydrological and infrastructure optimization tools through a combination of geospatial analytics and scenario generation algorithms. The underlying CWatM and MESSAGEix open source modeling tools could be interchanged with other similar tools in use by national and basin planning agencies. NEST incorporates detailed representation of the EWL sectors and linkages among them. The representation of these sectors builds mostly on open global data, facilitating transferability to other geographical regions and the definition of Basin-Country-Units (BCUs) embedding geopolitical borders. Among these data, we make use of 3-D cross-sectoral resource

flows and potentials, such as water availability, hydropower and renewable capacity. Additional local data can substitute or complement global data in empowering the model, facilitating calibration and validation and for building stakeholder trust.

The application of NEST to the Indus River Basin demonstrates the usefulness of such a tool in highlighting cross-sectoral policy impacts. An example are the implications of water treatment and recycling policies on energy consumption and the consequences for agriculture when attaining river environmental flow standards. Moreover, the delineation of the model into spatial units and the parametrization based on spatial data, enables results interrogation for single countries or BCUs within the basin boundaries. In this context, results for Pakistan and India are very different for water supply, electricity generation and agriculture.

Finally, critical areas for possible future improvement include: increasing spatial resolution and capability to deal with ultra-high resolution data; iterating MESSAGEix and CWatM to obtain a dynamic solution and better representing the non-linear interactions between groundwater and surface water; and, the improving assumptions at the geographical (and model) boundaries, for instance with cost curves or market models for food and electricity to represent the options of international trade.

*Code and data availability.* Code and processed data for NEST v1.0 is made available in Vinca (2020) (https://doi.org/10.5281/zenodo. 3625776) . The associated development repository with continuous updates can be found at https://github.com/iiasa/NEST .

The code and documentation for CWatM can also be found at: https://cwatm.iiasa.ac.at .

Documentation and code of MESSAGEix is available at: https://messageix.iiasa.ac.at .

*Author contributions.* AV, SiP, EB, VK, ZK and KR conceived of the framework. AV and SiP led model development, with support from EB, PB, ZK, VK, FD, YaW, AI, NR and AK. PB, YaW and YoW contributed the development and calibration of the hydrological model. IS and StP contributed algorithms and analysis for wind and solar systems representation. AV led analysis of results and preparation of figures, data tables and online repositories. All authors contributed to the review of the modeling framework and the writing of the manuscript. SiP coordinated the research.

*Competing interests.* The authors declare that they have no conflict of interest.

*Acknowledgements.* The authors acknowledge the Global Environment Facility (GEF) for funding the development of this research as part of the Integrated Solutions for Water, Energy, and Land (ISWEL) project (GEF Contract Agreement: 6993), and the support of the United Nations Industrial Development Organization (UNIDO). Part of this research was developed during the Young Scientists Summer Program at the International Institute for Applied Systems Analysis (IIASA), with financial support from the IIASA Annual Fund. The research has also been supported by the University of Victoria's Building Connections internal grant, the Natural Sciences and Engineering Research Council

of Canada. Internal support was provided by the Center for Water Informatics & Technology (WIT) at Lahore University of Management Sciences (LUMS), Pakistan and Coordenação de Aperfeiçoamento de Pessoal de Nível Superior - Brasil (CAPES). The hydrological model development was partly funded by the Belmont Forum Sustainable Urbanisation Global Initiative's Food-Water-Energy Nexus theme, for which coordination and research were supported by the US National Science Foundation under grant ICER/EAR-1829999 to Stanford University and by the Austrian Research Promotion Agency under the FUSE project funded to IIASA (grant agreement 730254 ).

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
