# Peer review of "The Nexus Solutions Tool (NEST) v1.0: An open platform for optimizing multi-scale energy-water-land system transformations"

_Geoscientific Model Development, 2019_

## Short Comment (SC1) · 26 Jul 2019

Dear authors,

in my role as Executive editor of GMD, I would like to bring to your attention our Editorial version 1.2:

https://www.geosci-model-dev.net/12/2215/2019/

This highlights some requirements of papers published in GMD, which is also available on the GMD website in the 'Manuscript Types' section: http://www.geoscientific-model-development.net/submission/manuscript_types.html

[Figure]

In particular, please note that for your paper, the following requirement has not been met in the Discussions paper:

- "The main paper must give the model name and version number (or other unique identifier) in the title."

Please provide the version number for NEST in the title of your revised manuscript.

Additionally, please note, that GMD is encouraging authors to provide a persistent access to the exact version of the source code used for the model version presented in the paper. As explained in https://www.geoscientific-model-development.net/about/manuscript_types.html the preferred reference to this release is through the use of a DOI which then can be cited in the paper. For projects in GitHub a DOI for a released code version can easily be created using Zenodo, see https://guides.github.com/activities/citable-code/ for details.

Finally note, that according to our new Editorial (v1.2) all data and analysis / plotting scripts should be made available.

Yours, Astrid Kerkweg

---

## Author Comment (AC1) · 29 Jul 2019

Dear Executive editor,

thank you for your comment. Following also the comment by the topical editor we have changed the title to include the version number. The updated manuscript is attached. In addition, we have updated the versioning information on the Github repository with code and data.

Thanks for suggesting Zenodo for publishing Github repositories. With other co-authors we are also considering other alternatives, but we will finalize a public repository with

[Figure]

DOI as soon as possible, also including the scripts used to generate figures.

As is does not seem possible to edit online title and repository links during the review stage, we will keep in mind to add the versioning number and updated links/DOIs to external repositories at the eventual finalization stage.

I hope with this we accomplish the suggested improvements. Sincerely, Adriano Vinca

Please also note the supplement to this comment:
https://www.geosci-model-dev-discuss.net/gmd-2019-134/gmd-2019-134-AC1-supplement.pdf

---

## Referee Comment (RC1) · Chrysi Laspidou (Referee) · 11 Aug 2019

The article includes an impressive line-up of authors from prestigious institutes around the world and proposes a model that is promising and will be appreciated by the scientific community. It presents a Nexus analysis, which is valuable and covers the synergies and trade-offs that are realized when a series of SDGs specifically considered are achieved. The case study of the Indus River Basin in South Asia is developed to showcase model results. This is a transboundary case study, including India and Pakistan, so it is really valuable to see how each country is affected differently.

Some of the issues with the paper that the authors should address are the following:

[Figure]

- It is not clear what scale the platform is developed for. My impression is that it is developed on a global scale, but only the Indus River Basin is presented in this article with the different SDG scenarios, so the scale is not very clear from the manuscript. This is important to mention in order to let the reader know what the potential of this platform is. - Data Sources: It is not clear where all the data presented in the article come from and how reliable it is. For example, in Figure 3 we see that for the water system the modelling entities are surface water, ground water and saline water, with each one of these streams being split to urban use, rural use, irrigation, etc. As a result, fresh water is allocated to a total of 12 "diversions", with many of these diversions having a different value for surface water and groundwater. Furthermore, urban and rural water use is split to piped and unpiped distribution and all kinds of waste streams are modelled separately. This is an impressively fine granularity of data, but how possible is it to define all this with some sort of reasonable accuracy for a country, or even a region? It is important that the authors show that this type of data is available and that it is reasonable to consider it in such a detail. Obviously, it is a plus to present the water system in great detail, but when the data cannot support that detail, then it becomes an important source of error. The fact that each one of the "diversions" has its corresponding energy associated with it—information that feeds the energy system—indicates that any error introduced in the water system with this classification will also be propagated to the energy system. If the authors use gross approximations for allocating demands to the different modules, then it is not clear how beneficial such detail is at the end. Experience shows that there is a lot of inaccuracy and error in this data and the modeller is better off relying on national or regional statistics, rather than on global databases. Whatever the case, the authors should definitely address this critical issue. Needless to say that the same issue of presenting an extreme level of detail without supporting it with the corresponding data applies for all systems, not only for the water system. The way it is presented right now, there is a serious lack of detailed explanation, which reduces the scientific reproducibility of the modelling science in the article. - SDG2 and Figure 8 (Land use by crop): Even though SDG2 refers

to food security, the authors do not clearly show how food security is affected. They show cotton and fodder that are not intended for human consumption, for example. Also, the fact that they show land used for crops and not yields makes a comparison difficult. For example, for the multiple SDG scenario, many of the crops are substituted by non-irrigated, which might have lower yields, when compared to irrigated. How does that affect food security? I understand that the authors explore SDG 2.4, which only refers to irrigation technologies, but it is a bit misleading when addressing SDG 2, to present and compare land used for crops and make no reference to food security and how food production is affected. - Figure 7(b): When comparing baseline and multiple SDG scenarios, we see that there is a great increase in Energy for Water and a great decrease in water for irrigation. But, how are Green House Gas emissions affected with such Energy increase and how is food security affected with such a reduction in irrigated land? This is an important question that comes to mind and is not addressed in the text. - Figure 7(a) / Nuclear Energy: I see that the authors drastically increase the use of nuclear energy in the suggested multiple SDG scenario. I assume that this was done due to the high efficiency of nuclear plants, which made possible to achieve the SDGs considered. However, it is not clear if such an increase in nuclear is desirable and/or even feasible for these countries. The amount of nuclear power used in the electricity mix of individual countries is a complex issue and it depends on many factors. It is not clear whether the authors have considered these factors for the case study presented, or whether the increase in nuclear power is merely a "modelling decision". - Discussion: What is missing from the manuscript is some discussion on the Nexus, in association with the results. For example, looking at Figure 7, when comparing baseline and multiple_SDG scenarios, we see that as one arrow gets thinner, another one gets thicker, which in a sense shows the effects of a Nexus analysis. In other words, we see the interdependence and "compromise" in resource use (we can't reduce everything at the same time, or as we reduce one sector, another one is affected). The choice of what is reduced and what is increased and the effects of these interlinkages is at the heart of a Nexus analysis and I feel that such a discussion after the presenta-

tion of results is missing. Also, the coherence of the SDGs themselves is relevant and should be discussed. How are things different when one tries to achieve only one goal vs. when multiple goals are considered. This is shown quite clearly in Figure 8, but the discussion on the coherence and/or conflicts of the SDGs themselves seems to be missing. - Uncertainty / Sensitivity: There is no mention of an uncertainty/sensitivity analysis of the results in the article. Such an assessment is necessary, even if it is limited, since in reality this uncertainty is multifaceted, involving human behaviour and is not so easily quantified.

---

## Referee Comment (RC2) · Anonymous Referee #2 · 11 Sep 2019

This article is well written and presents a substantive body or work. I enjoyed reading the paper and can see the value in the conclusions reached and thus the motivation of the research and sharing it with the scientific community. However, having read the paper, I find I'm missing various details that would greatly enhance my confidence in the conclusions, meaning some substantive modifications should be made prior to publication.

The introduction and description of the modelling framework were very clear. The presentation of some aspects of the model is left to other papers, however given the complexity of the model and focus on linking existing models this seemed a sensible

approach.

There is a brief presentation of hydrological model calibration and performance in section 2.3, but beyond this it is not very clear to me what the outcomes of the model are sensitive to and to what extend uncertainties is various parameters and components might impact upon the outcomes. The model is very complex and has many parameters, but what is it sensitive to in this test case. I would assume many of the components have a minor effect on the outcomes. The computation time was not clear to me, apologies if I have missed this, thus it's difficult to know what a realistic expectation for the authors is in this regard however at the very least this issue requires more discussion.

How the model was parameterised is also not very clear to me. The combination of tables 1 and 2 do not seem to represent all the data layers required by the model and they don't clearly (to me) map onto model parameters or distinct elements of the system. Perhaps this would be too long for the main text, but could it be a supplement? I'm not criticising the research as such but I don't feel I adequately understand the model data requirements from the text.

The limitations section is primarily a list of things that could be added to the model in future versions, in my opinion it's not sufficiently critical of the current model as implemented and the outputs. The text chooses to focus on several things that could be added without much evidence of how sensitive model results might be to these. There should be a discussion around the data sets needed, how well these can define model parameters and what implications these might have on the reliability of the conclusions.

Specific points: Figure 6: What simulations does this plot? Is it the mean of calibrated simulations by CWatM for the four climate models? Why not present the range and performance stats for each simulation? Section 2.3: Multiple climate models are used, but what about uncertainties in the other component? Why have an ensemble for this and then a deterministic set of parameters for the hydrological model? P24: "However,

it brings greater computational challenges associated with using classical mathematical programming methods" perhaps I missed it but what is the computational burden of the model and how does resolution affect this?

---

## Author Comment (AC2) · 11 Sep 2019

Thank you for your review of our paper. We are glad that you find the research valuable. In this comment we report the reviewer's comments under <> brackets, followed by our replies. attached is the updated version or the manuscript, where also new figures can be displayed.

< Some of the issues with the paper that the authors should address are the following:

- It is not clear what scale the platform is developed for. My impression is that it is developed on a global scale, but only the Indus River Basin is presented in this article

with the different SDG scenarios, so the scale is not very clear from the manuscript. This is important to mention in order to let the reader know what the potential of this platform is. >

The NEST framework has been so far implemented for the Indus River Basin. The data, come from a combination of global databases or global modelling assessments and local data sources which were collected specifically for the Indus Basin study. The global data is cropped to the basin-scale. The use of global data makes the implementation flexible in the sense that a first-cut analysis of the system can be made using available sources consistent across regions. We think that the two following statements explain it quite clearly, however we introduced an additional sentence to avoid ambiguity.

We state in line 13 pg 3 of the revised version that: "The new decision-making and open modeling platform provides a flexible framework for identifying and assessing EWL nexus solutions that can be applied to different geographic regions and multiple spatial and temporal scales." We also state in line 3 pg 8 of the revised version that: "To enable a transboundary perspective, the approach further intersects the sub-basin boundaries with country administrative units; sub-national administrative units and regions covering multiple basins (e.g., a country) could be considered."

To clarify this, we added this explanatory sentence in the manuscript: Line 32 pg 7: "The current framework focuses on an individual river basin. Future work will adapt the framework to expand and connect multiple basins."

<- Data Sources: It is not clear where all the data presented in the article come from and how reliable it is. For example, in Figure 3 we see that for the water system the modelling entities are surface water, ground water and saline water, with each one of these streams being split to urban use, rural use, irrigation, etc. As a result, fresh water is allocated to a total of 12 "diversions", with many of these diversions having a different value for surface water and groundwater. Furthermore, urban and rural water use is split to piped and unpiped distribution and all kinds of waste streams are

modelled separately. This is an impressively fine granularity of data, but how possible is it to define all this with some sort of reasonable accuracy for a country, or even a region? It is important that the authors show that this type of data is available and that it is reasonable to consider it in such a detail. Obviously, it is a plus to present the water system in great detail, but when the data cannot support that detail, then it becomes an important source of error. The fact that each one of the "diversions" has its corresponding energy associated with itâA ËŸTinformation that feeds the energy sys- ËĞ temâA ËŸTindicates that any error introduced in the water system with this classification ËĞ will also be propagated to the energy system. If the authors use gross approximations for allocating demands to the different modules, then it is not clear how beneficial such detail is at the end. Experience shows that there is a lot of inaccuracy and error in this data and the modeller is better off relying on national or regional statistics, rather than on global databases. Whatever the case, the authors should definitely address this critical issue. Needless to say that the same issue of presenting an extreme level of detail without supporting it with the corresponding data applies for all systems, not only for the water system. The way it is presented right now, there is a serious lack of detailed explanation, which reduces the scientific reproducibility of the modelling science in the article. >

Thanks for the comment, it help us realize that some aspects are not clearly described. Firstly, it is important to clarify that the purpose of the paper is not to perform a policy-relevant scenario analysis but instead to demonstrate the key features of the model and the interactions it enables users to investigate.

We now clearly write it in the text on line 28 page 15 that: "In this article we present an illustrative comparison between a baseline (business as usual) scenario and a multi-objective scenario achieving multiple SDG indicators by 2030. The analysis is not meant to provide a policy-relevant scenario analysis but instead to demonstrate the key features of the model and the interactions it enables users to investigate. Ongo-ing work is focusing on calibrating the model to local stakeholder perspectives and the

analysis of future pathways relevant for policy-makers. These results will be presented in a future publication." Secondly, concerning data we tried not refer to specific data sources we used for the Indus Basin, but rather clarify what type of data we linked in the framework. This is because we claim the NEST framework is flexible to easily change data sources in case of need. What is important, and you highlight it with your comment is to make the reader understand how the system described in the text and figures is linked with data. This lead to the main clarification to address the issue you raised. The boxes in the system diagrams represent the portfolio of possible technologies that the model can 'decide' to install and apply to supply the given demands. To give an example, if we see a certain amount of surface water used in piped distribution in 2020 (optimization time horizon 2020-2050), it is a solution of the optimization process that 'chooses' investments, capacity and utilization activity of these technologies. The only data that characterize these technologies are costs and consumptions per unit of production (parametric data). The data is collected from the literature and the databases outlined in the paper. This type of data collection and modeling is typical in the energy, water and land-use planning literature, and the integration of these planning frameworks is the main contribution of the model. Other historical data and projection assumptions are quite exhaustively described in the three sector sections.

However we have improved the text to explain the different types of data used, adding the following at the end of the Reference system Architecture subsection: "Two broad categories of data are used to characterize the NEST reference system: historical data on resource use and availability (synchronized with exogenous projections into future time horizon) and historical technology installed capacity; parametric data for technologies used in the optimization model, expressed as costs or consumption of resources per unit of production. These data are based on assumptions and can vary spatially or over time. The information is used in the optimization process to determine the model solution and output into cost and resource use trends that can then be compared to current situation. Figures 3, 4 and 5 show diagrams of the EWL sectors in the reference system. Some boxes represent resources availability or demands

characterized by historical data and future projections as explained in the following sections. Other boxes represent technologies parameterized with per unit costs and consumptions assumptions, for which specific data sources are reported in the case study section on the Indus River Basin."

Finally, the modeling framework is fully reproducible and available for download on github. Hence we refer to the model as 'open' in the title and throughout the text.

< - SDG2 and Figure 8 (Land use by crop): Even though SDG2 refers to food security, the authors do not clearly show how food security is affected. They show cotton and fodder that are not intended for human consumption, for example. Also, the fact that they show land used for crops and not yields makes a comparison difficult. For example, for the multiple SDG scenario, many of the crops are substituted by non-irrigated, which might have lower yields, when compared to irrigated. How does that affect food security? I understand that the authors explore SDG 2.4, which only refers to irrigation technologies, but it is a bit misleading when addressing SDG 2, to present and compare land used for crops and make no reference to food security and how food production is affected. >

Thanks for the observation. Concerning SDG 2 we only explore the production/supply aspect and its sustainability (regardless it is food crop or other). With the current framework we do not represent distributional food access for the population or nutrition rate. However, food demand is scaled with population projections equally in the baseline and in the multiple SDG scenario. We added the following text after Scenario the table and edited the table to avoid misunderstanding.

"Energy, water and agriculture yield demands are kept equal to the baseline scenario, which assumes SSP projections. Further work will aim to disentangle the distributional variation needed to increase equality in line with the SDG targets across different social groups (e.g. urban rural)."

< - Figure 7(b): When comparing baseline and multiple SDG scenarios, we see that

there is a great increase in Energy for Water and a great decrease in water for irrigation. But, how are Green House Gas emissions affected with such Energy increase and how is food security affected with such a reduction in irrigated land? This is an important question that comes to mind and is not addressed in the text. >

Thank you, we have improved the explanation of Figure 7b as below. To briefly answer your questions: in the multiple SDG scenario there is a stringent constraint on GHG emission. Therefore even if energy use in the water sector increases, the produces energy should mostly come from nuclear, hydropower or renewable energies (Figure 8);

Similarly, in the multiple SDG scenario, food demand (all crop production demand) is the same as in the baseline. What changes is the irrigation system, which allows to significantly save water from the irrigation sector to allocate it for environmental flows, or other uses.

Edits, line 25 pg 19 "This is to support increased water access in the municipal sector and massively expanded wastewater treatment capabilities in urban areas, but still represent less than 2\% of total energy production in 2020. Given GHG emission target set for the multiple SDG scenario, increase in energy production does not imply increase in emissions, as the generation fleet is mostly carbon neutral (See Figure 8). On the other hand, water withdrawals for agriculture reduce relative to the baseline scenario, while meeting the same crop production, due to the increase in rain-fed agriculture and more efficient irrigation."

< - Figure 7(a) / Nuclear Energy: I see that the authors drastically increase the use of nuclear energy in the suggested multiple SDG scenario. I assume that this was done due to the high efficiency of nuclear plants, which made possible to achieve the SDGs considered. However, it is not clear if such an increase in nuclear is desirable and/or even feasible for these countries. The amount of nuclear power used in the electricity mix of individual countries is a complex issue and it depends on many factors. It is not

clear whether the authors have considered these factors for the case study presented, or whether the increase in nuclear power is merely a "modelling decision".>

The choice of nuclear was merely a modelling decision, in later work on the basin we are refining constraints on Nuclear given it is a critical technology for the region.

As disclaimer we added the following sentence when describing SDG7 scenario: "Since nuclear is currently a critical issue in both India and Pakistan, further research will investigate the feasibility of nuclear with more detail and interacting with local stakeholders."

<- Discussion: What is missing from the manuscript is some discussion on the Nexus, in association with the results. For example, looking at Figure 7, when comparing baseline and multiple_SDG scenarios, we see that as one arrow gets thinner, another one gets thicker, which in a sense shows the effects of a Nexus analysis. In other words, we see the interdependence and "compromise" in resource use (we can't reduce everything at the same time, or as we reduce one sector, another one is affected). The choice of what is reduced and what is increased and the effects of these interlinkages is at the heart of a Nexus analysis and I feel that such a discussion after the presentation of results is missing. Also, the coherence of the SDGs themselves is relevant and should be discussed. How are things different when one tries to achieve only one goal vs. when multiple goals are considered. This is shown quite clearly in Figure 8, but the discussion on the coherence and/or conflicts of the SDGs themselves seems to be missing. >

Added the following paragraph after Figure 7 explanations: "These results demonstrate the value of interconnection across EWL sectors in terms of chain reaction in investments (i.e. expanding piping distribution also require expansion in electricity production and distribution), synergies (investing in irrigation efficiency implies saving in water distribution for irrigation) and trade-offs, as it is clearly not possible to minimize costs and resource use across all sectors to achieve the SDGs."

And, at the end of the paragraph on synergies among SDG: "Considering multiple target simultaneously shows different results than summing individual analysis. As mentioned above, the electricity mix changes when considering water constraints and climate targets. Similarly, land use is different when efficiency policies are in place together with environmental targets. This clearly shows the importance of an integrated multi-sectoral analysis to highlight synergies and barriers among objectives. The authors intend expand this topic in upcoming research."

<- Uncertainty / Sensitivity: There is no mention of an uncertainty/sensitivity analysis of the results in the article. Such an assessment is necessary, even if it is limited, since in reality this uncertainty is multifaceted, involving human behaviour and is not so easily quantified.>

We added a new paragraph with some sensitivity analysis driven by uncertainty in scenario assumptions (SSP, RCP and climate models). However we leave more parametric sensitivity analysis to coming publication that focus more on specific results. In this review process I also removed Figure 10 (b) that was representing the seasonality in crops, because it often lead to misunderstanding. I explained the concept in the text and also made room for the new Figure 11 that show the sensitivity analysis.

Please also note the supplement to this comment:
https://www.geosci-model-dev-discuss.net/gmd-2019-134/gmd-2019-134-AC2-supplement.pdf
* * *

---

## Author Comment (AC3) · 5 Nov 2019

Thank you for your review of our paper. We are glad that you find the research valuable. In this comment we report the reviewer's comments under <> brackets, followed by our replies. Attached is the updated version or the manuscript, where also some text has been re-shuffled and new figures and tables have been added.

< There is a brief presentation of hydrological model calibration and performance in section 2.3, but beyond this it is not very clear to me what the outcomes of the model are sensitive to and to what extend uncertainties is various parameters and components might impact upon the outcomes. The model is very complex and has many

parameters, but what is it sensitive to in this test case. I would assume many of the components have a minor effect on the outcomes. The computation time was not clear to me, apologies if I have missed this, thus it's difficult to know what a realistic expectation for the authors is in this regard however at the very least this issue requires more discussion. >

Thank you for the comments and for expressing the need of more uncertainty assessment. We added a section on sensitivity to major scenario assumptions, both for the outcomes of CWaTM and MESSAGEix. We decided not to include parametric sensitivity as our tests show lower uncertainty compared to SSP and RCP scenarios, and the number of parameters involved is very high. Future publication focusing on more precise sectorial questions will also explore the related parametric uncertainty.

< How the model was parameterised is also not very clear to me. The combination of tables 1 and 2 do not seem to represent all the data layers required by the model and they don't clearly (to me) map onto model parameters or distinct elements of the system. Perhaps this would be too long for the main text, but could it be a supplement? I'm not criticising the research as such but I don't feel I adequately understand the model data requirements from the text. >

Most of the update in this review expands the data section, in the main text and in the appendix. We expanded the summary table on data (Table1) to cover each single data source. We also included in the appendixes several tables on demand assumptions, historical conditions, costs and other technology parameters. An additional csv file will be attached to the SI, including data on solar and wind variable capacity factor.

< The limitations section is primarily a list of things that could be added to the model in future versions, in my opinion it's not sufficiently critical of the current model as implemented and the outputs. The text chooses to focus on several things that could be added without much evidence of how sensitive model results might be to these. There should be a discussion around the data sets needed, how well these can define model parameters and what implications these might have on the reliability of the conclusions.>

We made minor changes to the limitation sections, excluding not possible improvements that are not evident from what shown in the paper and adding some reference to results. We have not included discussion on the quality of the datasets used for the Indus case study as we would like to present the framework as applicable to other scopes and flexible in terms of data sources.

< Specific points: Figure 6: What simulations does this plot? Is it the mean of calibrated simulations by CWatM for the four climate models? Why not present the range and performance stats for each simulation? Section 2.3: Multiple climate models are used, but what about uncertainties in the other component? Why have an ensemble for this and then a deterministic set of parameters for the hydrological model? >

Yes, the plot shows the mean of calibrated simulations. It is a quite standard figure often shown in (https://www.geosci-model-dev-discuss.net/gmd-2019-214/). For sure including all climate model outputs would be interesting but also less clear to visualize. Concerning sensitivity, as mentioned in the first answer, we show the variability for different climate models.

< P24: "However, it brings greater computational challenges associated with using classical mathematical programming methods" perhaps I missed it but what is the computational burden of the model and how does resolution affect this? >

We added the following paragraph in the 'Model setup' section:

[revised manuscript text omitted]

**Appendix A: Exogenous demands**

[Figure]

**Figure A1.** Exogenous agriculture products (a), electricity (b) and water (c) demands. For each country and the whole Indus basin, from 2010 to 2060.

**Appendix B: Canals, hydroelectric projects and power plants data**

| Project | Capacity [ m$^3$ / sec ] | Length [ km ] | Long. In [ °E ] | Lat. In [ °N ] | Long. Out [ °E ] | Lat. Out [ °N |
|---|---|---|---|---|---|---|
| Rasul-Qadirabad | 538 | 44 | 73.5187 | 32.6830 | 73.7135 | 32.3370 |
| Qadirabad-Bulloki | 527 | 129 | 73.6858 | 32.3228 | 73.9138 | 31.2982 |
| Balloki-Sulemanki | 524 | 87 | 73.8590 | 31.2226 | 73.9241 | 30.4953 |
| Trimmu-Sidhnai | 312 | 64 | 72.1462 | 31.1450 | 72.1933 | 30.5690 |
| Sidhnai-Mailsi | 283 | 94 | 72.1582 | 30.5713 | 72.2459 | 29.7278 |
| Chashma-Jhelum | 615 | 135 | 71.3837 | 32.4358 | 72.2214 | 31.9680 |
| Taunsa-Panjnad | 340 | 72 | 70.8505 | 30.5137 | 71.3677 | 30.2735 |
| Marala-Ravi | 622 | 97 | 74.4698 | 32.6699 | 74.6239 | 31.8966 |
| Ravi-Bedian | 142 | 82 | 74.4701 | 31.7212 | 74.1755 | 30.7265 |
| Bambanwala-Ravi | 142 | 82 | 74.2941 | 32.3609 | 74.4701 | 31.7212 |
| Chenab-Bambanwala | 453 | 28 | 74.4698 | 32.6699 | 74.2941 | 32.3609 |
| Chenab-Ravi | 311 | 44 | 74.2941 | 32.3609 | 74.0820 | 31.4142 |
| Keenjhar-Karachi | 31 | 44 | 68.0500 | 24.9500 | - | - |
| Indira Gandhi | 138 | 602 | 75.0111 | 31.1628 | - | - |

**Table B1.** Major conveyance canals in the NEST implementation of the IRB that are linking river systems or for interbasin transfers. Locations, capacities and lengths are approximate and estimated by the authors based on reported technical data. Interbasin transfers are occurring where no outlet location is defined.

| Project | Country | Longitude [ °E ] | Latitude [ °N ] | Capacity [ MW ] | Storage [ km$^3$ ] | Opening |
|---|---|---|---|---|---|---|
| Azad Pattan | Pakistan | 73.5715 | 33.7678 | 700 | - | 2022 |
| Patrind | Pakistan | 73.4288 | 34.3440 | 150 | - | 2017 |
| Gulpur | Pakistan | 73.8625 | 33.4553 | 102 | - | 2019 |
| Suki Kinari | Pakistan | 73.5427 | 34.7231 | 870 | - | 2022 |
| Kohala | Pakistan | 73.6546 | 34.2023 | 1100 | - | 2025 |
| Athmuqam | Pakistan | 73.9107 | 34.5891 | 350 | - | 2020 |
| Golen Gol | Pakistan | 72.0143 | 35.9212 | 58 | - | 2018 |
| Mahl | Pakistan | 73.5667 | 34.9167 | 590 | - | 2025 |
| Neelum-Jhelum | Pakistan | 73.7189 | 34.3928 | 968 | - | 2018 |
| Diamer-Bhasha | Pakistan | 73.7370 | 35.5207 | 4500 | 10.5 | 2023 |
| Tarbela Extension | Pakistan | 72.6983 | 34.0897 | 1410 | - | 2018 |
| Karot | Pakistan | 73.6012 | 33.5998 | 720 | - | 2021 |
| Kalabagh | Pakistan | 71.6136 | 32.9564 | 3600 | 7.5 | proposed |
| Munda | Pakistan | 71.5330 | 34.3532 | 740 | 0.9 | proposed |
| Bunji | Pakistan | 74.6159 | 35.6358 | 7100 | 0.2 | proposed |
| Dasu | Pakistan | 73.1933 | 35.3173 | 4320 | 0.8 | 2021 |
| Akhori | Pakistan | 72.4528 | 33.6905 | 600 | 8.6 | 2025 |
| Sharmai | Pakistan | 72.0053 | 35.2766 | 150 | 0.3 | 2023 |
| Kishanganga | India | 74.7647 | 34.6475 | 360 | - | 2018 |
| Sawalkote | India | 75.0759 | 33.1691 | 1856 | - | proposed |
| Kirthai I | India | 75.1994 | 33.3868 | 390 | - | proposed |
| Kirthai II | India | 75.1994 | 33.3868 | 930 | - | proposed |
| Pakal Dul | India | 75.8136 | 33.4572 | 1000 | 0.1 | proposed |
| Kwar | India | 75.8280 | 33.3623 | 540 | - | proposed |
| Kiru | India | 75.8898 | 33.3518 | 624 | - | proposed |
| Bursar | India | 75.6956 | 33.3903 | 800 | 0.6 | proposed |
| Ujh | India | 75.5156 | 32.5590 | 212 | - | proposed |

**Table B2.** Additional planned hydropower projects included in the NEST implementation of the IRB. Locations, capacities and dates are approximate and estimated by the authors based on reported technical data.

[Figure]

**Figure B1.** Existing and planned power plant capacity in the NEST implementation of the IRB.

**Appendix C: Costs and Capacity Factor assumptions**

Tables C1 and C2 show costs and capacity factor values for a number of technologies in the model. Technologies like transmission lines or water canals are not included, as costs are dependent on the length and the region. The tables only include a subset of energy technologies with different cooling systems. Costs for gas plant in combined cycle ($cc$), single turbine ($gt$) and steam turbine ($st$) configurations are reported for all cooling systems, air cooling ($ac$), closed loop ($cl$) and once through cooling ($oc$). Although we model these different cooling systems also for oil, coal, geothermal and nuclear power plants, we only show costs for closed loop in Tables C1 and C2.

Solar and wind power plants are divided in three groups having same cost assumptions but different levels of capacity factor, attached in the Supplementary Information *Variable_capacity_factor.csv*.

| Technology | $I_{cost}$ [unit] | $F_{cost}$ [unit] | $Var_{cost}$ [unit] | CF [-] |
|---|---|---|---|---|
| Crops | [$/ha] | [$/ha] | [$/(ha month)] | |
| wheat | 341 | 36 | 72 | 1 |
| rice | 716 | 15 | 22 | 1 |
| cotton | 416 | 9 | 13 | 1 |
| fodder | 130 | 3 | 4 | 1 |
| sugarcane | 849.6 | 17 | 25.5 | 1 |
| pulses | 1320 | 26.4 | 39.6 | 1 |
| maize | 1000 | 20 | 30 | 1 |
| fruit | 545.5 | 11.5 | 16.5 | 1 |
| vegetables | 1362.5 | 27.5 | 41 | 1 |
| Energy Technolgies | [$/kW] | [$/kW] | [$/kWh] | |
| coal st cl | 6860 | 24 | 0.048611 | 0.9 |
| electricity distribution industry | 1120 | 36 | 0.034722 | 0.9 |
| electricity distribution irrigation | 1120 | 36 | 0.034722 | 0.9 |
| electricity distribution rural | 1120 | 36 | 0.034722 | 0.9 |
| electricity distribution urban | 1120 | 36 | 0.034722 | 0.9 |
| electricity short strg | 3000 | 16 | 0.020833 | 0.9 |
| gas cc ac | 1105 | 17 | 0.037264 | 0.9 |
| gas cc cl | 1064 | 16 | 0.036875 | 0.9 |
| gas cc ot | 1023 | 15 | 0.036111 | 0.9 |
| gas gt | 676 | 7 | 0.122222 | 0.9 |
| gas st cl | 1205 | 17 | 0.048611 | 0.9 |
| geothermal cl | 6343 | 135 | 0.025 | 0.9 |
| hydro old | 5000 | 15 | 0 | 0.95 |
| igcc cl | 4131 | 32 | 0.079167 | 0.9 |
| nuclear cl | 5751 | 97 | 0.029167 | 0.9 |
| oil cc cl | 1064 | 16 | 0.036875 | 0.9 |
| oil gt | 676 | 7 | 0.122222 | 0.9 |
| solar pv 1 | 3873 | 15 | 0.004167 | variable |
| solar pv 2 | 3873 | 15 | 0.004167 | variable |
| solar pv 3 | 3873 | 15 | 0.004167 | variable |
| wind 1 | 7000 | 8 | 0 | variable |
| wind 2 | 7000 | 8 | 0 | variable |
| wind 3 | 7000 | 8 | 0 | variable |

**Table C1.** Investment, fixed and variable costs and capacity factor values for model technologies

| Technology | $I_{cost}$ [unit] | $F_{cost}$ [unit] | $Var_{cost}$ [unit] | CF [-] |
|---|---|---|---|---|
| Irrigation technologies | [\$/ha] | [\$/ha] | [\$/(ha month)] | |
| drip | 2600 | 52 | 78 | 0.9 |
| flood | 460 | 10 | 14 | 0.9 |
| sprinkler | 1625 | 33 | 49 | 0.9 |
| canal lining flood | 3110 | 62 | 94 | 0.9 |
| smart | 2825 | 57 | 85 | 0.9 |
| drip smart | 3100 | 62 | 93 | 0.9 |
| sprinkler smart | 2125 | 43 | 64 | 0.9 |
| Water diversion/treatment technologies | [\$/(mq /day)] | [\$/(mq /day)] | | |
| industry gw diversion | 20 | 8.5 | 0 | 0.9 |
| industry sw diversion | 57 | 3 | 0 | 0.9 |
| industry wastewater recycling | 1350 | 99 | 0 | 0.9 |
| industry wastewater treatment | 431 | 37 | 0 | 0.9 |
| irrigation gw diversion | 8.5 | 1 | 0 | 0.9 |
| irrigation sw diversion | 57 | 3 | 0 | 0.9 |
| rural gw diversion | 8.5 | 1 | 0 | 0.9 |
| rural piped distribution | 326 | 18 | 0 | 0.9 |
| rural sw diversion | 57 | 3 | 0 | 0.9 |
| rural wastewater recycling | 1350 | 99 | 0 | 0.9 |
| rural wastewater treatment | 759 | 77 | 0 | 0.9 |
| smart irrigation sw diversion | 62.7 | 3.3 | 0 | 0.9 |
| urban gw diversion | 20 | 8.5 | 0 | 0.9 |
| urban piped distribution | 1013 | 252 | 0 | 0.9 |
| urban sw diversion | 57 | 3 | 0 | 0.9 |
| urban wastewater collection | 785 | 251 | 0 | 0.9 |
| urban wastewater irrigation | 1350 | 99 | 0 | 0.9 |
| urban wastewater recycling | 1350 | 99 | 0 | 0.9 |
| urban wastewater treatment | 431 | 37 | 0 | 0.9 |

**Table C2.** Investment, fixed and variable costs and capacity factor values for model technologies

[Figure]

**Figure C1.** Average highest irrigated and non-irrigated yield for each crop-country pairing in the NEST implementation of the IRB as well as the corresponding rates of residue generation.

[revised manuscript text omitted]

---

## Author Response (AR3)

Response to Review 1

Blue: Reviewer comments, black: Author response

< The article includes an impressive line-up of authors from prestigious institutes around
the world and proposes a model that is promising and will be appreciated by the scientific community. It
presents a Nexus analysis, which is valuable and covers the synergies and trade-offs that are realized when a
series of SDGs specifically considered are
achieved. The case study of the Indus River Basin in South Asia is developed to showcase model results.
This is a transboundary case study, including India and Pakistan,
so it is really valuable to see how each country is affected differently.>

Thank you for your review of our paper. We are glad that you find the research valuable.

< Some of the issues with the paper that the authors should address are the following:

- It is not clear what scale the platform is developed for. My impression is that it is
developed on a global scale, but only the Indus River Basin is presented in this article
with the different SDG scenarios, so the scale is not very clear from the manuscript.
This is important to mention in order to let the reader know what the potential of this
platform is. >

The NEST framework has been so far implemented for the Indus River Basin.
The data, come from a combination of global databases or global modelling assessments and local data
sources which were collected specifically for the Indus Basin study. The global data is cropped to the basin-scale. The use of global data makes the implementation flexible in the sense that a first-cut analysis of the
system can be made using available sources consistent across regions.

We think that the two following statements explain it quite clearly, however we introduced an additional
sentence to avoid ambiguity.

We state in line 13 pg 3 of the revised version that: "The new decision-making and open modeling platform
provides a flexible framework for identifying and assessing EWL nexus solutions that can be applied to
different geographic regions and multiple spatial and temporal scales."

We also state in line 3 pg 8 of the revised version that: "To enable a transboundary perspective, the approach
further intersects the sub-basin boundaries with country administrative units; sub-national administrative
units and regions covering multiple basins (e.g., a country) could be considered."

To clarify this, we added this explanatory sentence in the manuscript:

Line 32 pg 7: "The current framework focuses on an individual river basin. Future work will adapt the
framework to expand and connect multiple basins."

<- Data Sources: It is not clear where all the data presented in the article
come from and how reliable it is. For example, in Figure 3 we see that for the water
system the modelling entities are surface water, ground water and saline water, with
each one of these streams being split to urban use, rural use, irrigation, etc. As a result, fresh water is
allocated to a total of 12 "diversions", with many of these diversions
having a different value for surface water and groundwater. Furthermore, urban and
rural water use is split to piped and unpiped distribution and all kinds of waste streams
are modelled separately. This is an impressively fine granularity of data, but how possible is it to define all
this with some sort of reasonable accuracy for a country, or even
a region? It is important that the authors show that this type of data is available and
that it is reasonable to consider it in such a detail. Obviously, it is a plus to present
the water system in great detail, but when the data cannot support that detail, then it

becomes an important source of error. The fact that each one of the "diversions" has
its corresponding energy associated with itâA ̆Tinformation that feeds the energy sys- ̆
temâA ̆Tindicates that any error introduced in the water system with this classification ̆
will also be propagated to the energy system. If the authors use gross approximations
for allocating demands to the different modules, then it is not clear how beneficial such
detail is at the end. Experience shows that there is a lot of inaccuracy and error in
this data and the modeller is better off relying on national or regional statistics, rather
than on global databases. Whatever the case, the authors should definitely address
this critical issue. Needless to say that the same issue of presenting an extreme level
of detail without supporting it with the corresponding data applies for all systems, not
only for the water system. The way it is presented right now, there is a serious lack of
detailed explanation, which reduces the scientific reproducibility of the modelling science in the article. >

Thanks for the comment, it help us realize that some aspects are not clearly described.

Firstly, it is important to clarify that the purpose of the paper is not to perform a policy-relevant scenario analysis but instead to demonstrate the key features of the model and the interactions it enables users to investigate.

We now clearly write it in the text on line 28 page 15 that: "In this article we present an illustrative comparison between a baseline (business as usual) scenario and a multi-objective scenario achieving multiple SDG indicators by 2030. The analysis is not meant to provide a policy-relevant scenario analysis but instead to demonstrate the key features of the model and the interactions it enables users to investigate. Ongoing work is focusing on calibrating the model to local stakeholder perspectives and the analysis of future pathways relevant for policy-makers. These results will be presented in a future publication."

Secondly, concerning data we tried not refer to specific data sources we used for the Indus Basin, but rather clarify what type of data we linked in the framework. This is because we claim the NEST framework is flexible to easily change data sources in case of need. What is important, and you highlight it with your comment is to make the reader understand how the system described in the text and figures is linked with data.

This lead to the main clarification to address the issue you raised. The boxes in the system diagrams represent the portfolio of possible technologies that the model can 'decide' to install and apply to supply the given demands. To give an example, if we see a certain amount of surface water used in piped distribution in 2020 (optimization time horizon 2020-2050), it is a solution of the optimization process that 'chooses' investments, capacity and utilization activity of these technologies.
The only data that characterize these technologies are costs and consumptions per unit of production (parametric data). The data is collected from the literature and the databases outlined in the paper. This type of data collection and modeling is typical in the energy, water and land-use planning literature, and the integration of these planning frameworks is the main contribution of the model.

Other historical data and projection assumptions are quite exhaustively described in the three sector sections. However we have improved the text to explain the different types of data used, adding the following at the end of the Reference system Architecture subsection:

"Two broad categories of data are used to characterize the NEST reference system: historical data on resource use and availability (synchronized with exogenous projections into future time horizon) and historical technology installed capacity; parametric data for technologies used in the optimization model, expressed as costs or consumption of resources per unit of production. These data are based on assumptions and can vary spatially or over time. The information is used in the optimization process to determine the model solution and output into cost and resource use trends that can then be compared to current situation.

Figures 3, 4 and 5 show diagrams of the EWL sectors in the reference system. Some boxes represent resources availability or demands characterized by historical data and future projections as explained in the

following sections. Other boxes represent technologies parameterized with per unit costs and consumptions assumptions, for which specific data sources are reported in the case study section on the Indus River Basin."

Finally, the modeling framework is fully reproducible and available for download on github. Hence we refer to the model as 'open' in the title and throughout the text.

< - SDG2 and Figure 8 (Land use by crop): Even though SDG2 refers to food security, the authors do not clearly show how food security is affected. They
show cotton and fodder that are not intended for human consumption, for example.
Also, the fact that they show land used for crops and not yields makes a comparison
difficult. For example, for the multiple SDG scenario, many of the crops are substituted
by non-irrigated, which might have lower yields, when compared to irrigated. How does
that affect food security? I understand that the authors explore SDG 2.4, which only
refers to irrigation technologies, but it is a bit misleading when addressing SDG 2, to
present and compare land used for crops and make no reference to food security and
how food production is affected. >

Thanks for the observation.  Concerning SDG 2 we only explore the production/supply aspect and its sustainability (regardless it is food crop or other).
With the current framework we do not represent distributional food access for the population or nutrition rate. However, food demand is scaled with population projections equally in the baseline and in the multiple SDG scenario. We added the following text after Scenario the table and edited the table to avoid misunderstanding.

"Energy, water and agriculture yield demands are kept equal to the baseline scenario, which assumes SSP projections. Further work will aim to disentangle the distributional variation needed to increase equality in line with the SDG targets across different social groups (e.g. urban rural)."

< - Figure 7(b): When comparing baseline and multiple
SDG scenarios, we see that there is a great increase in Energy for Water and a great
decrease in water for irrigation. But, how are Green House Gas emissions affected
with such Energy increase and how is food security affected with such a reduction in
irrigated land? This is an important question that comes to mind and is not addressed
in the text. >

Thank you, we have improved the explanation of Figure 7b as below. To briefly answer your questions:
in the multiple SDG scenario there is a stringent constraint on GHG emission. Therefore even if energy use in the water sector increases, the produces energy should mostly come from nuclear, hydropower or renewable energies (Figure 8);
Similarly, in the multiple SDG scenario, food demand (all crop production demand) is the same as in the baseline. What changes is the irrigation system, which allows to significantly save water from the irrigation sector to allocate it for environmental flows, or other uses.

Edits, line 25 pg 19

"This is to support increased water access in the municipal sector and massively expanded wastewater treatment capabilities in urban areas, but still represent less than 2\% of total energy production in 2020. Given GHG emission target set for the multiple SDG scenario, increase in energy production does not imply increase in emissions, as the generation fleet is mostly carbon neutral (See Figure 8). On the other hand, water withdrawals for agriculture reduce relative to the baseline scenario, while meeting the same crop production, due to the increase in rain-fed agriculture and more efficient irrigation."

< - Figure 7(a) / Nuclear Energy: I see that the authors drastically increase the
use of nuclear energy in the suggested multiple SDG scenario. I assume that this was

done due to the high efficiency of nuclear plants, which made possible to achieve the
SDGs considered. However, it is not clear if such an increase in nuclear is desirable
and/or even feasible for these countries. The amount of nuclear power used in the electricity mix of
individual countries is a complex issue and it depends on many factors.
It is not clear whether the authors have considered these factors for the case study
presented, or whether the increase in nuclear power is merely a "modelling decision".>

The choice of nuclear was merely a modelling decision, in later work on the basin we are refining constraints on Nuclear given it is a critical technology for the region.

As disclaimer we added the following sentence when describing SDG7 scenario:

"Since nuclear is currently a critical issue in both India and Pakistan, further research will investigate the feasibility of nuclear with more detail and interacting with local stakeholders."

<- Discussion: What is missing from the manuscript is some discussion on the Nexus, in
association with the results. For example, looking at Figure 7, when comparing baseline and multiple_SDG
scenarios, we see that as one arrow gets thinner, another one
gets thicker, which in a sense shows the effects of a Nexus analysis. In other words,
we see the interdependence and "compromise" in resource use (we can't reduce everything at the same time,
or as we reduce one sector, another one is affected). The
choice of what is reduced and what is increased and the effects of these interlinkages
is at the heart of a Nexus analysis and I feel that such a discussion after the presenta
tion of results is missing. Also, the coherence of the SDGs themselves is relevant and
should be discussed. How are things different when one tries to achieve only one goal
vs. when multiple goals are considered. This is shown quite clearly in Figure 8, but
the discussion on the coherence and/or conflicts of the SDGs themselves seems to be
missing. >

Added the following paragraph after Figure 7 explanations:

"These results demonstrate the value of interconnection across EWL sectors in terms of chain reaction in investments (i.e. expanding piping distribution also require expansion in electricity production and distribution), synergies (investing in irrigation efficiency implies saving in water distribution for irrigation) and trade-offs, as it is clearly not possible to minimize costs and resource use across all sectors to achieve the SDGs."

And, at the end of the paragraph on synergies among SDG:

"Considering multiple target simultaneously shows different results than summing individual analysis. As mentioned above, the electricity mix changes when considering water constraints and climate targets. Similarly, land use is different when efficiency policies are in place together with environmental targets. This clearly shows the importance of an integrated multi-sectoral analysis to highlight synergies and barriers among objectives. The authors intend expand this topic in upcoming research."

<- Uncertainty / Sensitivity: There is no mention of an uncertainty/sensitivity
analysis of the results in the article. Such an assessment is necessary, even if it is
limited, since in reality this uncertainty is multifaceted, involving human behaviour and
is not so easily quantified.>

We added a new paragraph with some sensitivity analysis driven by uncertainty in scenario assumptions (SSP, RCP and climate models). However we leave more parametric sensitivity analysis to coming publication that focus more on specific results.

In this review process I also removed Figure 10 (b) that was representing the seasonality in crops, because it often lead to misunderstanding. I explained the concept in the text and also made room for the new Figure 11 that show the sensitivity analysis.

Review 2

This article is well written and presents a substantive body or work. I enjoyed reading the paper and can see the value in the conclusions reached and thus the motivation of the research and sharing it with the scientific community. However, having read the paper, I find I'm missing various details that would greatly enhance my confidence in the conclusions, meaning some substantive modifications should be made prior to publication.
The introduction and description of the modelling framework were very clear. The presentation of some aspects of the model is left to other papers, however given the complexity of the model and focus on linking existing models this seemed a sensible approach.
< There is a brief presentation of hydrological model calibration and performance in section 2.3, but beyond this it is not very clear to me what the outcomes of the model are sensitive to and to what extend uncertainties is various parameters and components might impact upon the outcomes.

The model is very complex and has many parameters, but what is it sensitive to in this test case. I would assume many of the components have a minor effect on the outcomes. The computation time was not clear to me, apologies if I have missed this, thus it's difficult to know what a realistic expectation for the authors is in this regard however at the very least this issue requires more discussion. >

Thank you for the comments and for expressing the need of more uncertainty assessment. We added a section on sensitivity to major scenario assumptions, both for the outcomes of CWaTM and MESSAGEix. We decided not to include parametric sensitivity as our tests show lower uncertainty compared to SSP and RCP scenarios, and the number of parameters involved is very high. Future publication focusing on more precise sectorial questions will also explore the related parametric uncertainty.

< How the model was parameterised is also not very clear to me. The combination of tables 1 and 2 do not seem to represent all the data layers required by the model and they don't clearly (to me) map onto model parameters or distinct elements of the system. Perhaps this would be too long for the main text, but could it be a supplement? I'm not criticising the research as such but I don't feel I adequately understand the model data requirements from the text. >

Most of the update in this review expands the data section, in the main text and in the appendix. We expanded the summary table on data (Table1) to cover each single data source.

We also included several tables in the SI on demand assumptions, historical conditions, costs and other technology parameters. An additional xls file will be attached to the SI, including data on solar and wind variable capacity factor.

< The limitations section is primarily a list of things that could be added to the model in future versions, in my opinion it's not sufficiently critical of the current model as implemented and the outputs. The text chooses to focus on several things that could be added without much evidence of how sensitive model results might be to these. There

should be a discussion around the data sets needed, how well these can define model parameters and what implications these might have on the reliability of the conclusions.>

We made minor changes to the limitation sections, excluding not possible improvements that are not evident from what shown in the paper and adding some reference to results.
We have not included discussion on the quality of the datasets used for the Indus case study as we would like to present the framework as applicable to other scopes and flexible in terms of data sources.

< Specific points: Figure 6: What simulations does this plot? Is it the mean of calibrated simulations by CWatM for the four climate models? Why not present the range and performance stats for each simulation? Section 2.3: Multiple climate models are used, but what about uncertainties in the other component? Why have an ensemble for this and then a deterministic set of parameters for the hydrological model? >

Yes, the plot shows the mean of calibrated simulations. It is a quite standard figure often shown in (https://www.geosci-model-dev-discuss.net/gmd-2019-214/). For sure including all climate model outputs would be interesting but also less clear to visualize.
Concerning sensitivity, as mentioned in the first answer, we show the variability for different climate models.

< P24: "However, it brings greater computational challenges associated with using classical mathematical programming methods" perhaps I missed it but what is the computational burden of the model and how does resolution affect this? >

We added the following paragraph in the 'Model setup' section

[revised manuscript text omitted]

**Appendix A:**

**Appendix A:** **Exogenous demands**

[Figure]

**Figure A1.** Exogenous agriculture products (a), electricity (b) and water (c) demands. For each country and the whole Indus basin, from 2010 to 2060.

**Appendix B: Canals, hydroelectric projects and power plants data**

| Project | Capacity [ m$^3$ / sec ] | Length [ km ] | Long. In [ °E ] | Lat. In [ °N ] | Long. Out [ °E ] | Lat. Out [ °N ] |
|---|---|---|---|---|---|---|
| Rasul-Qadirabad | 538 | 44 | 73.5187 | 32.6830 | 73.7135 | 32.3370 |
| Qadirabad-Bulloki | 527 | 129 | 73.6858 | 32.3228 | 73.9138 | 31.2982 |
| Balloki-Sulemanki | 524 | 87 | 73.8590 | 31.2226 | 73.9241 | 30.4953 |
| Trimmu-Sidhnai | 312 | 64 | 72.1462 | 31.1450 | 72.1933 | 30.5690 |
| Sidhnai-Mailsi | 283 | 94 | 72.1582 | 30.5713 | 72.2459 | 29.7278 |
| Chashma-Jhelum | 615 | 135 | 71.3837 | 32.4358 | 72.2214 | 31.9680 |
| Taunsa-Panjnad | 340 | 72 | 70.8505 | 30.5137 | 71.3677 | 30.2735 |
| Marala-Ravi | 622 | 97 | 74.4698 | 32.6699 | 74.6239 | 31.8966 |
| Ravi-Bedian | 142 | 82 | 74.4701 | 31.7212 | 74.1755 | 30.7265 |
| Bambanwala-Ravi | 142 | 82 | 74.2941 | 32.3609 | 74.4701 | 31.7212 |
| Chenab-Bambanwala | 453 | 28 | 74.4698 | 32.6699 | 74.2941 | 32.3609 |
| Chenab-Ravi | 311 | 44 | 74.2941 | 32.3609 | 74.0820 | 31.4142 |
| Keenjhar-Karachi | 31 | 44 | 68.0500 | 24.9500 | - | - |
| Indira Gandhi | 138 | 602 | 75.0111 | 31.1628 | - | - |

**Table B1.** Major conveyance canals in the NEST implementation of the IRB that are linking river systems or for interbasin transfers. Locations, capacities and lengths are approximate and estimated by the authors based on reported technical data. Interbasin transfers are occurring where no outlet location is defined.

| Project | Country | Longitude [ °E ] | Latitude [ °N ] | Capacity [ MW ] | Storage [ km³ ] | Opening |
|---------|---------|------------------|-----------------|-----------------|-----------------|---------|
| Azad Pattan | Pakistan | 73.5715 | 33.7678 | 700 | - | 2022 |
| Patrind | Pakistan | 73.4288 | 34.3440 | 150 | - | 2017 |
| Gulpur | Pakistan | 73.8625 | 33.4553 | 102 | - | 2019 |
| Suki Kinari | Pakistan | 73.5427 | 34.7231 | 870 | - | 2022 |
| Kohala | Pakistan | 73.6546 | 34.2023 | 1100 | - | 2025 |
| Athmuqam | Pakistan | 73.9107 | 34.5891 | 350 | - | 2020 |
| Golen Gol | Pakistan | 72.0143 | 35.9212 | 58 | - | 2018 |
| Mahl | Pakistan | 73.5667 | 34.9167 | 590 | - | 2025 |
| Neelum-Jhelum | Pakistan | 73.7189 | 34.3928 | 968 | - | 2018 |
| Diamer-Bhasha | Pakistan | 73.7370 | 35.5207 | 4500 | 10.5 | 2023 |
| Tarbela Extension | Pakistan | 72.6983 | 34.0897 | 1410 | - | 2018 |
| Karot | Pakistan | 73.6012 | 33.5998 | 720 | - | 2021 |
| Kalabagh | Pakistan | 71.6136 | 32.9564 | 3600 | 7.5 | proposed |
| Munda | Pakistan | 71.5330 | 34.3532 | 740 | 0.9 | proposed |
| Bunji | Pakistan | 74.6159 | 35.6358 | 7100 | 0.2 | proposed |
| Dasu | Pakistan | 73.1933 | 35.3173 | 4320 | 0.8 | 2021 |
| Akhori | Pakistan | 72.4528 | 33.6905 | 600 | 8.6 | 2025 |
| Sharmai | Pakistan | 72.0053 | 35.2766 | 150 | 0.3 | 2023 |
| Kishanganga | India | 74.7647 | 34.6475 | 360 | - | 2018 |
| Sawalkote | India | 75.0759 | 33.1691 | 1856 | - | proposed |
| Kirthai I | India | 75.1994 | 33.3868 | 390 | - | proposed |
| Kirthai II | India | 75.1994 | 33.3868 | 930 | - | proposed |
| Pakal Dul | India | 75.8136 | 33.4572 | 1000 | 0.1 | proposed |
| Kwar | India | 75.8280 | 33.3623 | 540 | - | proposed |
| Kiru | India | 75.8898 | 33.3518 | 624 | - | proposed |
| Bursar | India | 75.6956 | 33.3903 | 800 | 0.6 | proposed |
| Ujh | India | 75.5156 | 32.5590 | 212 | - | proposed |

**Table B2.** Additional planned hydropower projects included in the NEST implementation of the IRB. Locations, capacities and dates are approximate and estimated by the authors based on reported technical data.

[Figure]

**Figure B1.** Existing and planned power plant capacity in the NEST implementation of the IRB.

**Appendix C: Costs and Capacity Factor assumptions**

Tables C1 and C2 show costs and capacity factor values for a number of technologies in the model. Technologies like transmission lines or water canals are not included, as costs are dependent on the length and the region. The tables only include a subset of energy technologies with different cooling systems. Costs for gas plant in combined cycle (*cc*), single turbine (*gt*) and steam turbine (*st*) configurations are reported for all cooling systems, air cooling (*ac*), closed loop (*cl*) and once through cooling (*oc*). Although we model these different cooling systems also for oil, coal, geothermal and nuclear power plants, we only show costs for closed loop in Tables C1 and C2.

Solar and wind power plants are divided in three groups having same cost assumptions but different levels of capacity factor, attached in the Supplementary Information *Variable_capacity_factor.csv*.

Parameter Value Snow melt coefficient 0.003597 Crop factor correction 1.211 Ice melt coefficient 0.5366 Soil preferential flow constant 5.4 ARNO b 1.259 Interflow part of recharge factor 1.807 Groundwater recession coefficient factor 3.823 Runoff concentration factor 1.492 Routing Manning's N 8.104 Reservoir normal storage limit 0.5257 Lake alpha factor 1.154 Lake wind factor 1.205

| Technology | $I_{cost}$ [unit] | $F_{cost}$ [unit] | $Var_{cost}$ [unit] | CF [-] |
|---|---|---|---|---|
| Crops | [$/ha] | [$/ha] | [$/(ha month)] | |
| wheat | 341 | 36 | 72 | 1 |
| rice | 716 | 15 | 22 | 1 |
| cotton | 416 | 9 | 13 | 1 |
| fodder | 130 | 3 | 4 | 1 |
| sugarcane | 849.6 | 17 | 25.5 | 1 |
| pulses | 1320 | 26.4 | 39.6 | 1 |
| maize | 1000 | 20 | 30 | 1 |
| fruit | 545.5 | 11.5 | 16.5 | 1 |
| vegetables | 1362.5 | 27.5 | 41 | 1 |
| Energy Technolgies | [$/kW] | [$/kW] | [$/kWh] | |
| coal st cl | 6860 | 24 | 0.048611 | 0.9 |
| electricity distribution industry | 1120 | 36 | 0.034722 | 0.9 |
| electricity distribution irrigation | 1120 | 36 | 0.034722 | 0.9 |
| electricity distribution rural | 1120 | 36 | 0.034722 | 0.9 |
| electricity distribution urban | 1120 | 36 | 0.034722 | 0.9 |
| electricity short strg | 3000 | 16 | 0.020833 | 0.9 |
| gas cc ac | 1105 | 17 | 0.037264 | 0.9 |
| gas cc cl | 1064 | 16 | 0.036875 | 0.9 |
| gas cc ot | 1023 | 15 | 0.036111 | 0.9 |
| gas gt | 676 | 7 | 0.122222 | 0.9 |
| gas st cl | 1205 | 17 | 0.048611 | 0.9 |
| geothermal cl | 6343 | 135 | 0.025 | 0.9 |
| hydro old | 5000 | 15 | 0 | 0.95 |
| igcc cl | 4131 | 32 | 0.079167 | 0.9 |
| nuclear cl | 5751 | 97 | 0.029167 | 0.9 |
| oil cc cl | 1064 | 16 | 0.036875 | 0.9 |
| oil gt | 676 | 7 | 0.122222 | 0.9 |
| solar pv 1 | 3873 | 15 | 0.004167 | variable |
| solar pv 2 | 3873 | 15 | 0.004167 | variable |
| solar pv 3 | 3873 | 15 | 0.004167 | variable |
| wind 1 | 7000 | 8 | 0 | variable |
| wind 2 | 7000 | 8 | 0 | variable |
| wind 3 | 7000 | 8 | 0 | variable |

**Table C1.** Calibration parameters Investment, fixed and variable costs and capacity factor values for convergence of CWatM model technologies

| Technology | $I_{cost}$ [unit] | $F_{cost}$ [unit] | $Var_{cost}$ [unit] | CF [-] |
|---|---|---|---|---|
| Irrigation technologies | [$/ha] | [$/ha] | [$/(ha month)] | |
| drip | 2600 | 52 | 78 | 0.9 |
| flood | 460 | 10 | 14 | 0.9 |
| sprinkler | 1625 | 33 | 49 | 0.9 |
| canal lining flood | 3110 | 62 | 94 | 0.9 |
| smart | 2825 | 57 | 85 | 0.9 |
| drip smart | 3100 | 62 | 93 | 0.9 |
| sprinkler smart | 2125 | 43 | 64 | 0.9 |
| Water diversion/treatment technologies | [$/(mq /day)] | [$/(mq /day)] | | |
| industry gw diversion | 20 | 8.5 | 0 | 0.9 |
| industry sw diversion | 57 | 3 | 0 | 0.9 |
| industry wastewater recycling | 1350 | 99 | 0 | 0.9 |
| industry wastewater treatment | 431 | 37 | 0 | 0.9 |
| irrigation gw diversion | 8.5 | 1 | 0 | 0.9 |
| irrigation sw diversion | 57 | 3 | 0 | 0.9 |
| rural gw diversion | 8.5 | 1 | 0 | 0.9 |
| rural piped distribution | 326 | 18 | 0 | 0.9 |
| rural sw diversion | 57 | 3 | 0 | 0.9 |
| rural wastewater recycling | 1350 | 99 | 0 | 0.9 |
| rural wastewater treatment | 759 | 77 | 0 | 0.9 |
| smart irrigation sw diversion | 62.7 | 3.3 | 0 | 0.9 |
| urban gw diversion | 20 | 8.5 | 0 | 0.9 |
| urban piped distribution | 1013 | 252 | 0 | 0.9 |
| urban sw diversion | 57 | 3 | 0 | 0.9 |
| urban wastewater collection | 785 | 251 | 0 | 0.9 |
| urban wastewater irrigation | 1350 | 99 | 0 | 0.9 |
| urban wastewater recycling | 1350 | 99 | 0 | 0.9 |
| urban wastewater treatment | 431 | 37 | 0 | 0.9 |

**Table C2.** Investment, fixed and variable costs and capacity factor values for model technologies

[Figure]

**Figure C1.** Average highest irrigated and non-irrigated yield for each crop-country pairing in the NEST implementation of the IRB as well as the corresponding rates of residue generation.

[revised manuscript text omitted]

**Editor's review**

Dear Adriano,

Thank you for your revised version of the manuscript. I believe it is much improved from the original version and would like to thank the reviewers for their constructive comments in the discussion. In my view the paper is an important contribution and thus should be published.

However, in line with the reviewers I do have concerns about the number of parameters in such a complex model, small changes to some of these may have profound impact on the outcomes while others may have negligible impact. I fully appreciate that a substantive sensitivity analysis is beyond the scope of this paper, it's not the purpose of the paper in my view, but I think in your response to the reviewers and edits to the manuscript this is perhaps underplayed given the level reviewers concerns.

Could I ask specifically that you review the sections outlined below as a technical revision.

Best wishes,
Jeff

Section 3.1 on model calibration.

This section is much improved however the section on calibration needs to be far more upfront on the limitations of the calibration conducted here. A distributed model with 13 parameters over a huge area has been calibrated to a single point. Thus, the model is likely to perform poorly in many other parts of the basin I would assume. At the moment, the difficulty of simulating the basin is highlighted but not the limitations of the calibration adopted – in a real application of the model more calibration data would be needed I assume? I don't think you need to change the calibration, but please be upfront about the limitations in this example. Furthermore, one-gauge location seems very limited, was this the only data available or did you chose one location for another reason?

4.3 Uncertainty section
Should this include model structure as a source of uncertainty? Presumably in such a complex system feedbacks and parts of the system could be omitted from the model, distorting the response other parts of the systems?

Around page 33 "We therefore leave data source uncertainty analysis to future publication that will to focus specifically on numerical outputs and implications" – Both reviews highlight this as a critical issues. I can appreciate that you are keen not to extend the paper to include a sensitivity analysis on the model parameters and data sources, but I think the reviewers raise legitimate concerns about the use of such a complex model for scenario analysis over large basins. The sentences here are insufficient at reflecting this in my opinion.

Thus, I'd be more comfortable if you either refute the opinion of the reviewers (and mine) or set out some expectations about how potentially challenging model parameterisation might be. My assumption is that data source and parametric uncertainty will likely have profound impact on the outcomes of the model and if you don't think this will be the case you need to explain why in the text.

Also please check the typo in this sentence.

Finally, I agree with Review 2 that the section on limitations is more focused on further development, or simply known limitations in model structure. Could you make the section title more specific in this regard?

**Author's response**

Dear Jeff,

Thank you for your comments and the revision work you did. I made further changes in response to the issues you rose.

In section 3.1 I now wrote more clearly that having one single station is a limitation for the hydrological model calibration. At page 16, line 18 of the newly uploaded manuscript I added:

"The Besham station is chosen because of its coverage of historical years, it incorporates the runoff from both glacial and seasonal snowmelt. However, multiple stations would be necessary to better represent regional heterogeneity (in particular lower versus upper basin). Future work will incorporate spatially distributed observations to improve the calibration. "

To address the concerns on the uncertainty, I added a preliminary parametric sensitivity analysis where input parameters are varied within a fixed range and we look at output variations.
An entire new paragraph is added at page 28, line 19.

At the end of this paragraph I also mention the importance of structural uncertainty, which I believe is substantial but it is also a too vast topic to be assessed in this article.

Finally, I changed the title of the limitations section.

I think that these changes, in particular the additional sensitivity analysis, further improve the accuracy of some sections. I hope the article is now better in line with your suggestions and the journal requirements.

Kind regards,
Adriano